# SpiderSolver: A Geometry-Aware Transformer for Solving PDEs on Complex Geometries

**Kai Qi**[1,*]**, Fan Wang**[1,*]**, Zhewen Dong**[1]**, Jian Sun**[1,2] (✉)

[1]School of Mathematics and Statistics, Xi'an Jiaotong University, Xi'an, China
[2]State Industry-Education Integration Center for Medical Innovations at Xi'an Jiaotong University
{qikai1218,wangfan525,dongzhewen}@stu.xjtu.edu.cn, jiansun@xjtu.edu.cn

## Abstract

Transformers have demonstrated effectiveness in solving partial differential equations (PDEs). However, extending them to solve PDEs on complex geometries remains a challenge. In this work, we propose SpiderSolver, a geometry-aware transformer that introduces spiderweb tokenization for handling complex domain geometry and irregularly discretized points. Our method partitions the irregular spatial domain into spiderweb-like patches, guided by the domain boundary geometry. SpiderSolver leverages a coarse-grained attention mechanism to capture global interactions across spiderweb tokens and a fine-grained attention mechanism to refine feature interactions between the domain boundary and its neighboring interior points. We evaluate SpiderSolver on PDEs with diverse domain geometries across seven datasets, including cars, airfoils, blood flow in the human thoracic aorta, as well as canonical cases governed by the Navier-Stokes, Darcy flow, elasticity, and plasticity equations. Experimental results demonstrate that SpiderSolver consistently achieves state-of-the-art performance across different datasets and metrics, with better generalization ability in the OOD setting. The code is available at https://github.com/Kai-Qi/SpiderSolver.

## 1 Introduction

Solving partial differential equations (PDEs) is fundamental to many computational problems in science and engineering. Classical numerical methods involve discretizing computational domains and solving the resulting algebraic systems. However, for domains with complex boundary geometries and irregular discretization, these methods require complex mesh generation and incur high computational costs. For example, computing the drag force on a car (Figure 1) requires solving the Navier-Stokes equations with a car-shaped boundary. Classical methods discretize the computational domain into irregular meshes or points (e.g., over 30,000 points for the Shape-Net Car dataset), leading to high computational complexity and cost.

In recent years, there has been growing interest in applying deep learning methods to solve PDEs, such as PINNs [1], neural operators [2, 3, 4], etc. However, most of the current methods deal with regular computational domains. For example, the well-established FNO [3] solves PDE in rectangular domains with uniform grids due to its implementation via the Fast Fourier Transform. To efficiently solve PDEs with irregularly shaped domains, Geometry-Informed Neural Operator (GINO) [5] and Geo-FNO [6] are proposed based on deforming the irregularly shaped domain into a uniform latent mesh. Such approaches encounter difficulties with mesh transformations for complex geometries of the computational domains. The Graph Neural Operator (GNO) [2] adopts the Graph Neural Network [7] as a backbone by formulating the kernel integral operator as message passing on graphs,

---

\* Equal contribution.

39th Conference on Neural Information Processing Systems (NeurIPS 2025).

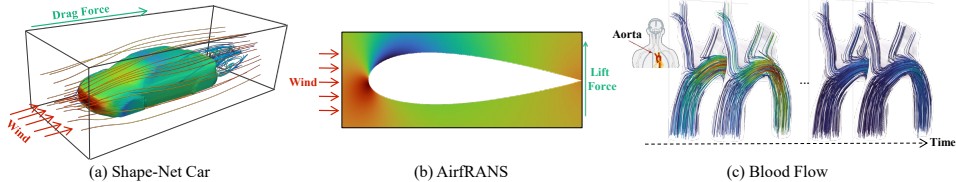

(a) Shape-Net Car               (b) AirfRANS               (c) Blood Flow

Figure 1: Visualization of datasets. For Shape-Net Car and AirfRANS, we estimate the surface pressure on the car/airfoil and the surrounding air velocity based on their shapes, aiming to predict the drag and lift forces for a driving car or a flying airplane. The object surfaces of cars, airfoils, and blood vessels serve as PDE domain boundaries, with the interior regions surrounding the car/airfoil surfaces or lying inside the blood vessel.

which can be directly applied to irregular meshes. 3D-GenCA [8] introduces a pre-trained 3D vision model [9] for GNO, to encode the complex boundary of the domain. However, it does not fully consider the complex physical relationships between boundary and interior mesh points.

Recently, Transformers [10], a widely used backbone in deep learning, has been applied to solve PDEs, which can also be directly applied on irregular meshes. FactFormer [11], MINO [12], and Galerkin Transformer [13] are based on the idea that the dot-product attention can be considered as an approximation of an integral transform with a non-symmetric learnable kernel function, which relates Transformers to the FNO. These transformers have demonstrated promising performance for solving PDEs with complex geometry. However, massive mesh points lead to the huge computational overhead of Transformers because the canonical attention in Transformers has quadratic complexity, and brings challenges for Transformers to capture the complex physical relations between irregular mesh points. Transolver [14] proposes a Physics-Attention mechanism that decomposes the discretized domain into a series of learnable slices, with mesh points exhibiting similar physical states assigned to the same slice. Notably, the slices are learned without explicitly utilizing the geometric information of the domain.

In this paper, we focus on designing a transformer-based PDE solver for domains with complex boundary geometries and irregularly discretized points. The main challenges include efficient tokenization over irregular computational domains and the integration of domain and boundary geometry into the network design. Following this motivation, we propose SpiderSolver, a geometry-aware Transformer. SpiderSolver uses spiderweb tokenization to partition the domain into spiderweb-like patches, guided by spectral clustering of the boundary and the distance of interior points from the boundary. Spiderweb tokenization partitions physical space in a more fundamental and physically intuitive way, achieving a trade-off between computational efficiency and capturing the physical interactions of spatial points. Based on spiderweb tokenization, as shown in Figure 2, SpiderSolver integrates coarse-grained and fine-grained attention to capture the physical relationship between points of interior domain and boundary surface. We evaluate our SpiderSolver on two industrial-level design tasks and one blood simulation task, as well as canonical cases governed by the Navier-Stokes and Darcy flow equations. These tasks are challenging since they require the model to handle various complex boundary geometries. The experiments show that our SpiderSolver achieved state-of-the-art results compared to other neural operators and Transformer-based PDE solvers for these tasks.

Overall, our contributions are as follows. *First*, we introduce a novel complex computational domain partitioning method, Spiderweb Tokenization, which divides the domain into spiderweb-like patches. *Second*, we propose SpiderSolver, a geometry-aware transformer for solving PDEs with complex boundary geometries and discretization. SpiderSolver integrates coarse-grained and fine-grained attention to capture the physical relationship between inner space and boundary surfaces. *Third*, SpiderSolver surpasses the state-of-the-art methods in the PDE solving tasks for car and airfoil design, the blood flow dynamics in the human thoracic aorta, as well as two fundamental PDE tasks.

## 2   Related Work

The deep learning methods have been widely applied to solve PDEs, which can be roughly categorized into two paradigms. The first paradigm, such as PINNs [1], Deep Ritz [15], etc, is to approximate the solutions of PDEs by neural networks and formalize the physical constraints (including equations,

initial and boundary conditions) as objective functions to optimize network parameters. This kind of method requires the exact formalization of PDEs and needs to retrain the network for new PDEs. Another paradigm is operator learning, e.g., DeepONet [4] and FNO [3]. It is to learn the nonlinear mapping (i.e., operator) between the function space of parameters and the function space of the PDEs' solutions. FNO [3] approximates the solution operator by the kernel integral operator and calculates it in the frequency domain by the fast Fourier transform (FFT). Afterward, a series of variants of FNO are proposed. F-FNO [16] enhances model efficiency by employing factorization in the Fourier domain. U-NO [17] and U-FNO [18] combine FNO and U-Net for multiscale problems. However, most of these methods solve PDEs with regular computational domains.

For solving PDEs with irregularly shaped domains, Geo-FNO [6] and GINO [5] are based on the idea of deforming the irregular input domain into a uniform latent mesh on which the FFT used in FNO can be applied. However, such mesh transformations are difficult for complex geometries of the computational domains, such as a car-shaped domain, resulting in performance degradation. Graph Neural Operator (GNO) [2] formulates the FNO's kernel integral operator as message passing on graphs, leveraging Graph Neural Networks [7] to construct neural operators. However, it should be noted that graph kernels are insufficient in their capacity to capture global information.

Transformers [10] have also been introduced as a backbone to solve PDEs with irregularly shaped domains. MINO [12], FactFormer [11], and Galerkin Transformer [13] are based on the idea that the dot-product attention can be considered as an approximation of an integral transform with a non-symmetric learnable kernel function, which relates Transformer to the FNO. Vito [19] combines the vision transformer [20] and the U-net [21] to construct the neural operator. To overcome the quadratic complexity of attention, GNOT [22] and ONO [23] utilize the well-established linear Transformers, such as Reformer [24], Performer [25]. These methods directly apply attention to mesh points, which is computationally prohibited when the number of points is large. Transolver [14] decomposes the discretized domain into a series of learnable slices, in which mesh points under similar physical states are assigned to the same slice. Then Transolver applies attention to these learnable slices to learning intrinsic physical relations.

In contrast, SpiderSolver is a geometry-aware Transformer-based PDE solver, which is designed to partition the domain into spiderweb-like patches utilizing physical and geometric knowledge to reduce the computation cost of attention. Besides, SpiderSolver adopts the multi-grained attention mechanism to capture the intricate physical correlation of complex boundaries and interior points.

## 3 Our Proposed SpiderSolver

**Problem setup.** We consider the partial differential equations (PDEs) defined over a domain $\Omega \in \mathbb{R}^d$, where $d$ represents the dimensionality of the space. In this work, the domain $\Omega$ has general irregular geometry and boundary, typically discretized into $N$ mesh points, represented as $G \in \mathbb{R}^{N \times d}$. Specifically, $G = \{g_i\}_{i=1}^{N_I} \cup \{s_j\}_{j=1}^{N_B}$, where $\mathcal{I}_G = \{g_i\}_{i=1}^{N_I}$ represents the $N_I$ interior points of $\Omega$ (off-boundary points), and $\mathcal{B}_G = \{s_j\}_{j=1}^{N_B}$ denotes the $N_B$ boundary points of $\Omega$, and $N_B + N_I = N$. Our goal is to learn a non-linear operator that outputs the physical field $u$ to approximate physical quantities over the geometry $G$.

For PDEs with domain boundaries in general irregular geometries (e.g., the boundaries shown in Figure 1), the physical quantities at different locations within the domain are influenced by their proximity to the boundary and the geometry of the boundary itself. In automotive aerodynamics, the flow field exhibits stratification around the vehicle. Near the car surface, airflow adheres to the surface geometry, while at greater distances, the velocity gradually aligns with the freestream. The aerodynamic effect is influenced by the car's surface curvature and angle of attack. Horizontal surfaces, such as the roof, have less impact on the flow, whereas regions with larger angles, such as the front windshield, significantly alter both the flow direction and speed.

Based on the above observations, we propose SpiderSolver, a geometry-aware transformer, specifically defined as a fast PDE solver for solving PDEs over the domains with irregular geometries. As shown in Figure 2, SpiderSolver integrates *spiderweb tokenization* (Figure 2 (a)) into a transformer architecture. As will be presented in Section 3.1, spiderweb tokenization partitions the domain of PDE into spiderweb-like patches considering the geometry of the domain boundary, and each patch is used to define a token. Built upon the spiderweb tokenization, we design a transformer (detailed in Section

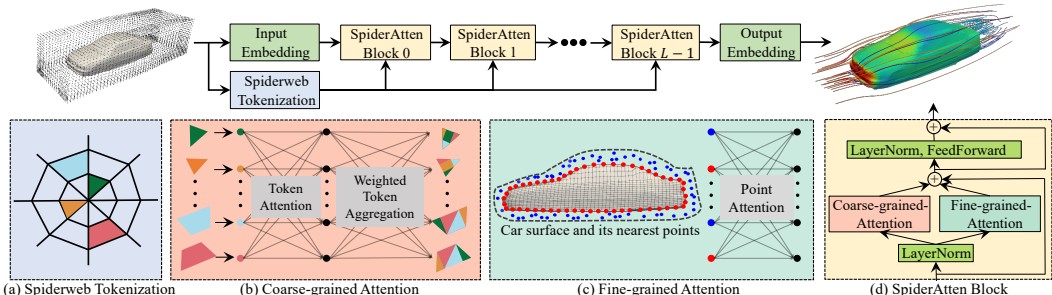

(a) Spiderweb Tokenization  (b) Coarse-grained Attention  (c) Fine-grained Attention  (d) SpiderAtten Block

Figure 2: Overall architecture of SpiderSolver. Spiderweb Tokenization partitions the domain into Spiderweb-like patches. Coarse-grained Attention interacts the features over the spiderweb tokens. Weighted Token Aggregation updates the point-wise features by the weighted combination of token features. Fine-grained Attention enables interactions between points of boundary and near boundary.

3.2) consisting of cascaded attention blocks, and each block is composed of a *coarse-grained attention* (Figure 2 (b)) over the spiderweb tokens and a *fine-grained attention* (Figure 2 (c)) to interact features between points of domain boundary and their near points in the interior of domain. This transformer is the basis of our SpiderSolver, which is learned to output the PDE solutions.

## 3.1  Spiderweb Tokenization over PDE Domain

**Overview.**  To solve PDEs with complex boundary geometries, we introduce a geometry-aware tokenization method as a foundational step in constructing our transformer. The aim is to quantize the computational domain $\Omega$ with interior point set $\mathcal{I}_G$ and boundary point set $\mathcal{B}_G$ into non-overlapping sub-regions adaptive to the geometry of domain boundary (e.g., the object surfaces in Figure 1). This process begins with spectral clustering for the domain boundary $\partial\Omega$ to quantize the domain boundary to sub-regions. Then the inner space of the domain $\Omega$ is partitioned based on the boundary clustering and the distance of interior points to the boundary. We term this approach "spiderweb tokenization" due to the spiderweb-like structure of the resulting divided sub-regions, as illustrated in Figure 3.

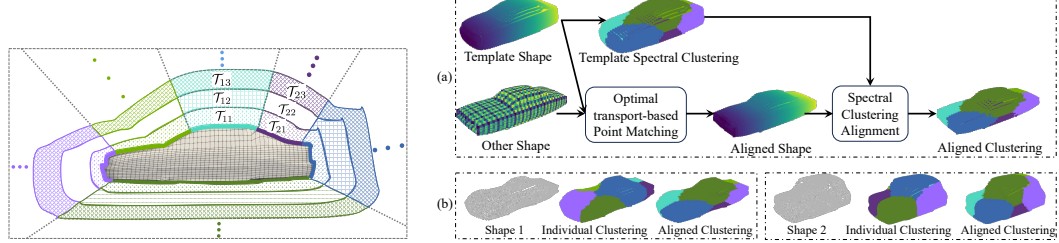

Figure 3: Spiderweb-like sub-region structure from spiderweb tokenization. Colored curves on the car surface illustrate spectral clustering-based partition.

Figure 4: Optimal transport-based alignment for spectral clustering. (a) Aligning surface points of a shape to the template shape for clustering. (b) Comparison of individual and aligned clustering results.

**Boundary spectral clustering.**  Spectral clustering over the boundary leverages the eigenvalue problem of the Laplace operator $\Delta$, which can be defined on non-Euclidean geometry for capturing its geometric and topological properties [26]. For the continuous domain of boundary surface $\partial\Omega \subset \mathbb{R}^h$, the eigenvalue problem is:

$$-\Delta u = \lambda u, \quad \text{in } \partial\Omega, \tag{1}$$

subject to appropriate boundary conditions. The eigenvalues $\lambda$ and eigenfunctions $u$ characterize the intrinsic geometric structure of the boundary $\partial\Omega$, enabling its decomposition into geometric decomposed sub-regions. These eigenvectors $u$ are used to embed the data into a low-dimensional space. Specifically, the Laplacian operator can be approximated by constructing the affinity matrix

over nearest neighbors (10 neighbors) graph using points or mesh vertices on the domain boundary. Then we use k-means clustering on the eigenvectors of the normalized Laplacian matrix associated with the $m_B$ smallest eigenvalues, to group $\partial\Omega$ into $m_B$ clusters. Spectral clustering leverages the graph structure of surface points to capture global, shape-aware connectivity beyond local geometry. See Appendix A for implementation details of spectral clustering.

**Aligned boundary clustering for different instances of object surface.** As shown in Figure 4 (b) and Figure 10 (Appendix B), individual clustering of different instances of an object results in unaligned clustering. We devise an aligned clustering approach by transferring the clustering of a template shape to the remaining instances of the boundary shape. As shown in Figure 4 (a), we first align all other shapes to a randomly selected reference shape using the entropic regularization optimal transport problem solved by the Sinkhorn algorithm [27]. We then average these aligned shapes to obtain a template shape, on which spectral clustering is performed. Finally, the clustering labels from the template shape are transferred to the other instances of the object. Here, the reference shape is only used for alignment, while the template shape serves as the basis for spectral clustering. Therefore, given a new instance of object, we only need to align its points to template shape points to derive its point clustering. This above strategy to align PDE boundaries to the boundary template shape can accomplish the varying shapes of boundary geometry for a given type of geometry, e.g., car, airfoil. Please refer to Figure 9 in Appendix B for the example of template shape.

**Spiderweb tokenization.** Based on aligned boundary clustering, we further divide the inner space of $\Omega$ into sub-regions $\mathcal{T}_{pq}$ with $p = 1, 2, \ldots, m_B$ and $q = 1, 2, \ldots, m_I$, as shown in Figures 3 and 2 (a). Spectral clustering applied to the boundary $\mathcal{B}_G$ yields $m_B$ clusters, therefore $\mathcal{B}_G = \bigcup_{p=1}^{m_B} \mathcal{C}_p$ with $\mathcal{C}_p$ denoting the $p$-th cluster of the boundary. As shown in Figure 3, interior PDE domain is divided into sub-regions, and each sub-region $\mathcal{T}_{pq}$ includes the interior point with its closest point on domain boundary belonging to $p$-th boundary cluster, and its signed distance function (SDF) values to domain boundary within a range $(d_{q-1}, d_q]$, with $d_0 = 0$. This region is expressed as:

$$\mathcal{T}_{pq} = \left\{ g \in \mathcal{I}_G \mid \text{SDF}(g) \in (d_{q-1}, d_q], \ s_j^* \in \mathcal{C}_p, s_j^* = \underset{s_j \in \mathcal{B}_G}{\arg\min} \|g - s_j\| \right\}. \tag{2}$$

We determine the range of $(d_{q-1}, d_q]$ by ensuring an equal number of points falling within each interval $(d_{q-1}, d_q]$. In this way, we partition the whole PDE domain into $M = m_B m_I + m_B$ sub-regions with $m_B m_I$ sub-regions in the interior domain and $m_B$ sub-regions on the boundary. Each sub-region is taken as the domain of a token.

**Notes:** The spiderweb-tokenization method is inherently geometry-aware, as explained below. *Firstly*, it utilizes the geometric information of boundary to partition the domain effectively. *Secondly*, by optimal transport-based point alignment, it ensures that the clustering of object surfaces, i.e., domain boundaries, remains consistent across different instances. This alignment enables a well-structured tokenization of PDE domain, even when boundary shapes vary. As a result, the geometrically aligned tokens allow transformers to function as PDE solvers that inherently account for boundary geometry.

### 3.2 Transformer Design for SpiderSolver

Based on spiderweb tokenization (Figure 2 (a)), we define SpiderSolver (Figure 2) as a transformer defined over spiderweb tokenization, with *SpiderAtten layers* integrating coarse-grained (Figure 2 (b)) and fine-grained attentions (Figure 2 (c)). The coarse-grained attention facilitates interactions among spiderweb-like tokens. The fine-grained attention serves as a finer-level complement, capturing interactions between boundary points and their near interior points adjacent to the boundary.

As shown in Figure 2, we summarize the overall process of SpiderSolver. First, a linear embedding maps the input $G \in \mathbb{R}^{N \times d}$ or with the observed physical quantity, to initial feature $\mathbf{x}^0 \in \mathbb{R}^{N \times C}$. Next, $\mathbf{x}^0$ passes sequentially through the $L$ SpiderAtten blocks (Figure 2 (d)). The $l$-th ($l \in [0, L-1]$) SpiderAtten block is defined as:

$$\mathbf{f}^l = \text{Spiderweb-Tokenization}\left(\mathbf{x}^l\right),$$
$$\tilde{\mathbf{x}}^{l+1} = \mathbf{x}^l + \text{T2P}\left(\text{Coarse-AT}\left(\mathbf{f}^l\right), \mathbf{x}^l\right) + \text{Fine-AT}\left(\text{LN}\left(\mathbf{x}^l\right)\right), \tag{3}$$
$$\mathbf{x}^{l+1} = \text{FeedForward}\left(\text{LN}\left(\tilde{\mathbf{x}}^{l+1}\right)\right) + \tilde{\mathbf{x}}^{l+1},$$

where the "Coarse-AT" refers to coarse-grained attention, "Fine-AT" refers to fine-grained attention, "LN" denotes layerNorm, and $\mathbf{x}^l, \tilde{\mathbf{x}}^{l+1} \in \mathbb{R}^{N \times C}, \mathbf{f}^l \in \mathbb{R}^{M \times C}$. The T2P operator transforms the

token features back to update the point-wise features. The "FeedForward" operator consists of two linear layers with a non-linear activation in between. The operation Spiderweb-Tokenization($\mathbf{x}^l$) is defined as the concatenation of average pooled features of $\mathbf{x}^l$ over spiderweb-token domains $\mathcal{T}_{pq}$, for $p \in [1, m_B], q \in [1, m_I]$, together with $m_B$ additional feature by average feature pooling over $m_B$ boundary clusters. This coarse-level tokenization facilitates the fast self-attention computation from the interactions of $N$ points to $M$ tokens.

After $L$ SpiderAtten blocks, a linear embedding layer over $\mathbf{x}^L$ is applied to obtain the output physical quantities, i.e., the PDE solution. We next introduce the operators of Coarse-AT($\cdot$), T2P($\cdot,\cdot$), Fine-AT($\cdot$) of Equation (3).

### 3.2.1 Coarse-grained Attention (Coarse-AT)

The coarse-grained attention (Figure 2 (b)), denoted as Coarse-AT in Equation (3), is defined over the $M$ tokens of the spiderweb tokenization of the PDE domain. For the $l$-th SpiderAtten block, with the token-level feature $\mathbf{f}^l \in \mathbb{R}^{M \times C}$, we apply the self attention, denoted as $\mathcal{A}(\cdot)$:

$$\hat{\mathbf{f}}^l = \mathcal{A}(\mathbf{f}^l) = [\mathcal{A}_1(\mathbf{f}^l); \cdots ; \mathcal{A}_h(\mathbf{f}^l)]W_0^l, \quad \text{where } \mathcal{A}_i(\mathbf{f}^l) = \text{Softmax}\left(Q_i K_i^\top / \sqrt{d_k}\right) V_i, \quad (4)$$

to conduct multi-head attention, where $d_k$ denotes the dimension of key vectors, $[\cdot]$ denotes concatenation, $T_i = \text{Linear}(\mathbf{f}^l, W_{T,i}^l)$ for $T_i \in \{Q_i, K_i, V_i\}, i = 1, \cdots, h$, and $h$ denotes the number of heads. "Linear" refers to a linear layer and generates $Q, K, V, \in \mathbb{R}^{M \times C}$ in attention. Note that, the self-attention is conducted over the spiderweb tokens instead of all the interior points of PDE domain. Thus, the output of the operator Coarse-AT is $\hat{\mathbf{f}}^l$, i.e., the coarse-grained attention feature.

### 3.2.2 From Token to Point-Wise Features (T2P)

After coarse-grained attention, the attended token features $\hat{\mathbf{f}}^l = \text{Coarse-AT}(\mathbf{f}^l)$ will be transformed into point-wise features and added to the corresponding point-wise features $\mathbf{x}^l$. Let $\mathbf{x}_I^l \in \mathbb{R}^{N_I \times C}$, $\mathbf{x}_B^l \in \mathbb{R}^{N_B \times C}$ be the point-wise features $\mathbf{x}^l$ in the interior and on the domain boundary respectively. Let $\hat{\mathbf{f}}_I^l \in \mathbb{R}^{m_B m_I \times C}$ and $\hat{\mathbf{f}}_B^l \in \mathbb{R}^{m_B \times C}$ be the spiderweb token features $\hat{\mathbf{f}}^l$ of the interior and boundary tokens respectively. The updated point-wise features are computed based on the *weighted token aggregation*, which globally aggregates all the token features to compute the point-wise update, formulated as:

$$\hat{\mathbf{x}}_I^l = \eta^l \hat{\mathbf{f}}_I^l, \hat{\mathbf{x}}_B^l = \xi_B^l \hat{\mathbf{f}}_B^l + \xi_I^l \hat{\mathbf{f}}_I^l, \text{ where } \eta^l = \text{Linear}(\mathbf{x}_I^l), \xi_B^l = \text{Linear}(\mathbf{x}_B^l), \xi_I^l = \text{Linear}(\mathbf{x}_I^l), \quad (5)$$

where the three "Linear" layers are respectively with learnable parameters $W_I^l, W_B^l$ and $W_{I2B}^l$, and generates $\eta^l \in \mathbb{R}^{N_I \times m_B m_I}, \xi_B^l \in \mathbb{R}^{N_B \times m_B}, \xi_I^l \in \mathbb{R}^{N_B \times m_B m_I}$ as weighting matrix to combine spiderweb token features. Therefore, the operator T2P outputs updated features $[\hat{\mathbf{x}}_I^l; \hat{\mathbf{x}}_B^l]$. It facilitates the point-wise feature updating by the coarse-grained attention features over spiderweb tokens.

### 3.2.3 Fine-grained attention (Fine-AT)

The fine-grained attention (Figure 2 (c)) is designed to enhance the interaction between features of points located on boundary and points near boundary. This design is motivated by the observation that as points get closer to boundary, their physical characteristics are increasingly influenced by the boundary geometry. In practical applications, crucial physical quantities, such as the drag coefficient of a car and the lift coefficient of an airfoil are derived from the features on or near the boundary.

To effectively capture these boundary-related interactions, fine-grained attention is introduced to facilitate feature interactions between the domain boundary points $\mathcal{B}_G$ and their nearest interior domain neighboring points in the set of $\mathcal{T}_B = \{g \in \mathcal{I}_G | \text{SDF}(g) \in (d_0, d_1)\}$. The corresponding features of these boundary and near-boundary interior points in $\mathcal{T}_B$ are concatenated to form a combined feature representation $\mathbf{x}_F$. The attention is then applied to this representation to capture the underlying physical dependencies by $\mathcal{A}(\mathbf{x}_F)$, which is *the output of the operator* Fine-AT.

### 3.2.4 Network Training and Testing

The proposed transformer contains the learnable parameters in the two embedding layers and $L$ SpiderAtten blocks in Equation (3). The network is trained based on the relative $L_2$-norm loss

between the network output and the ground-truth PDE solutions over the training set [14]. In the training phase, spiderweb tokenization is performed over the training dataset as a data pre-processing procedure. In the testing phase, given a new instance of the PDE geometric domain and/or boundary condition, the PDE domain is divided into token sub-regions by aligned spectral clustering, and then the SpiderSolver outputs the PDE solution based on the spiderweb tokenization.

## 4 Experiments

We evaluate SpiderSolver on five datasets spanning industrial, biomedical, and fundamental PDE tasks. The template shapes of Shape-Net Car and AirfRANS are visualized in Figure 9 in Appendix B. As the Blood Flow, Bounded Navier-Stokes and Darcy Flow datasets have fixed geometries, optimal transport-based alignment is not required. See Appendix C for more details on five datasets.

**Shape-Net Car** [33] consists of 889 simulated samples based on Reynolds-Average Navier-Stokes equations, with car shapes from the ShapeNet "car" category [34]. Each sample includes velocity and pressure fields solved via a finite element method over 32,186 mesh points. Using irregularly discretized car and surrounding space as input, the model is trained to predict these fields, from which the drag coefficient is subsequently derived. The shapes of cars randomly selected from Shape-Net Car are visualized in Figure 7 in Appendix B.

Table 1: Results on Shape-Net Car and AirfRANS datasets. Vol: error of surrounding physics field; Surf: error of surface physics field. $C_D$, $C_L$: error of drag and lift coefficients; $\rho_D$, $\rho_L$: Spearman's rank correlation of drag and lift coefficients.

| Methods | Shape-Net Car | | | | AirfRANS | | | |
|---|---|---|---|---|---|---|---|---|
| | Vol↓ | Surf↓ | $C_D$↓ | $\rho_D$↑ | Vol↓ | Surf↓ | $C_L$↓ | $\rho_L$↑ |
| Simple MLP | 0.0512 | 0.1304 | 0.0307 | 0.9496 | 0.0081 | 0.0200 | 0.2108 | 0.9932 |
| G-SAGE [28] | 0.0461 | 0.1050 | 0.0270 | 0.9695 | 0.0087 | 0.0184 | 0.1476 | 0.9964 |
| PointNet [29] | 0.0494 | 0.1104 | 0.0298 | 0.9583 | 0.0253 | 0.0996 | 0.1973 | 0.9949 |
| G-U-Net [30] | 0.0471 | 0.1102 | 0.0226 | 0.9725 | 0.0076 | 0.0146 | 0.1677 | 0.9944 |
| MG-Net [31] | 0.0354 | 0.0781 | 0.0168 | 0.9840 | 0.0214 | 0.0387 | 0.2252 | 0.9945 |
| GNO [2] | 0.0383 | 0.0815 | 0.0172 | 0.9834 | 0.0269 | 0.0405 | 0.2016 | 0.9934 |
| Galerkin [13] | 0.0339 | 0.0878 | 0.0179 | 0.9764 | 0.0074 | 0.0159 | 0.2336 | 0.9957 |
| Geo-FNO [6] | 0.1670 | 0.2378 | 0.0664 | 0.8280 | 0.0361 | 0.0820 | 0.6614 | 0.9257 |
| GNOT [32] | 0.0329 | 0.0798 | 0.0178 | 0.9833 | 0.0049 | 0.0152 | 0.1992 | 0.9942 |
| GINO [5] | 0.0386 | 0.0810 | 0.0184 | 0.9826 | 0.0297 | 0.0482 | 0.1821 | 0.9958 |
| 3D-GeoCA [8] | 0.0319 | 0.0779 | 0.0159 | 0.9842 | / | / | / | / |
| Transolver [14] | 0.0228 | 0.0793 | 0.0129 | 0.9916 | 0.0025 | 0.0080 | 0.1026 | 0.9977 |
| **SpiderSolver** | **0.0210** | **0.0738** | **0.0100** | **0.9928** | **0.0017** | **0.0043** | **0.0741** | **0.9988** |

**AirfRANS** [35] comprises 1,000 high-fidelity simulations over 32,000-point meshes of 4- and 5-digit NACA airfoils, with variations in shape, Reynolds number, and angle of attack. Using irregularly discretized airfoil and surrounding space as input, the model is trained to predict velocity, pressure, and viscosity fields, from which lift coefficient is computed. The shapes of airfoils are visualized in Figure 8 in Appendix B.

**Blood Flow** dataset [36] consists of 500 simulations of blood flow in a fixed human thoracic aorta geometry, by the Navier-Stokes equations. Each sample varies in inlet and outlet conditions, and is discretized over a tetrahedral mesh with 1,656 spatial nodes and 121 time steps. The model is trained to predict the velocity field of the flow, using pressure at the inlets and outlets.

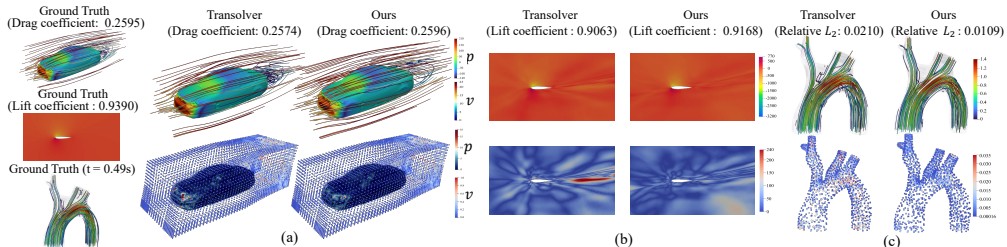

Figure 5: Examples from three datasets. The first column shows the ground truth. In (a), (b), and (c), the first row presents the predictions by Transolver and our model, while the second row displays the point-wise $L_2$ norm of the difference between ground truth and predictions.

**Bounded Navier-Stokes dataset with multiple separate boundaries** [37] simulates 2D fluid flow through a pipe containing several fixed pillar-like obstacles, resulting in multiple disconnected boundaries. Despite the presence of separate boundaries, the SDF at each point is uniquely defined as the minimum distance to all boundary components. See Appendix C for more details on datasets.

**Darcy Flow dataset** [3] considers the steady-state of the 2-d Darcy Flow equation on the unit box. In the Darcy Flow dataset, we take the spatial outer rectangle boundary as the boundary to compute clustering and SDF for tokenization. See Appendix C for more details on datasets.

**Elasticity and plasticity datasets.** Elasticity [6] models a unit cell with an arbitrary void, governed by the incompressible Rivlin–Saunders constitutive law. Plasticity [6] considers a block impacted by a rigid die, governed by an elasto-plastic constitutive model with time-dependent deformation.

**Implementation details.** To ensure that our model parameters are comparable to other Transformer-based models, such as Transolver, we set the number of layers as 8 and the channel of hidden features as 256 or 512, depending on the number of observed quantities of input data. All experiments are performed on a GeForce RTX 4090 GPU. See Appendix C for more details.

Table 2: Results on Blood Flow dataset.

| Methods | Velocity $\downarrow$ |
|---|---|
| Simple MLP | 0.3080 |
| DeepONet [4] | 0.8926 |
| POD-D [38] | 0.3742 |
| Geo-FNO [6] | 0.1209 |
| GNOT [32] | 0.0411 |
| NORM [36] | 0.0453 |
| Geo-FNO [6] | 0.1209 |
| 3D-GeoCA [8] | 0.2863 |
| GINO [5] | 0.1864 |
| Transolver [14] | 0.0438 |
| **SpiderSolver** | **0.0322** |

**Compared methods and metrics.** We compare SpiderSolver with more than 18 baselines. For the Shape-Net Car and AirfRANS, we evaluate the estimation error of physical fields in the PDE domain (Vol), domain boundary surface (Surf), drag coefficient ($C_D$) and lift coefficient ($C_L$) using the relative $L_2$ except that we follow [35, 14] to use MSE for the Vol and Surf on the AirfRANS dataset. We additionally use Spearman's rank correlations of drag and lift coefficients, respectively in the Shape-Net Car and AirfRANS datasets. For the other datasets, we use the relative $L_2$ error as the metric. See Appendix E for details.

## 4.1 Main Results

**Results on Shape-Net Car.** As shown in Table 1, SpiderSolver outperforms various methods. The Spearman's rank correlation indicates that our predicted drag coefficients better match the true ranking. Figure 11 in Appendix B shows an example of spiderweb tokenization by SpiderSolver.

**Results on AirfRANS.** As shown in Table 1, SpiderSolver reduces the MSE of the volume and surface physics fields by $32.0\%$ and $46.3\%$, compared to Transolver. Figures 12 and 13 in Appendix B show an example of spiderweb tokenization by SpiderSolver and the patches learned by Transolver.

**Results on Blood Flow.** As shown in Table 2, SpiderSolver achieves state-of-the-art performance. Figure 5 (c) visualizes the predicted velocity fields.

**Results on Bounded Navier-Stokes and Darcy Flow.** SpiderSolver outperforms various methods on both Bounded Navier-Stokes and Darcy flow datasets. See Tables 13 and 14 in Appendix F for results. Darcy Flow dataset is defined on a regular quadrilateral domain, and both the Bounded Navier-Stokes and Darcy Flow datasets use uniform grids. Although SpiderSolver is designed for complex boundaries and irregular grids, it remains applicable to such settings. Figure 14 in Appendix B shows two examples of spiderweb tokenization on Bounded Navier-Stokes dataset. For the Bounded Navier–Stokes dataset, SpiderSolver applies spectral clustering to all obstacle surfaces, where coarse-grained attention directly captures in-

Table 3: Generalization in OOD on AirfRANS.

| Methods | Reynolds OOD | | Angles OOD | |
|---|---|---|---|---|
| | $C_L \downarrow$ | $\rho_L \uparrow$ | $C_L \downarrow$ | $\rho_L \uparrow$ |
| Simple MLP | 0.6205 | 0.9578 | 0.4128 | 0.9572 |
| G-SAGE [28] | 0.4333 | 0.9707 | 0.2538 | 0.9894 |
| PointNet [29] | 0.3836 | 0.9806 | 0.4425 | 0.9784 |
| G-U-Net [30] | 0.4664 | 0.9645 | 0.3756 | 0.9186 |
| GNO [2] | 0.4408 | 0.9878 | 0.3038 | 0.9836 |
| Galerkin [13] | 0.4615 | 0.9862 | 0.3814 | 0.9821 |
| GNOT [32] | 0.3268 | 0.9865 | 0.3497 | 0.9863 |
| GINO [5] | 0.4180 | 0.9645 | 0.2583 | 0.9923 |
| Transolver [14] | 0.3889 | 0.9911 | 0.2490 | 0.9940 |
| **SpiderSolver** | **0.2291** | **0.9922** | **0.1062** | **0.9941** |

teractions among tokens. As shown in Figure 14 (Appendix B), the resulting interior partitions align well with obstacle geometry, enabling attention maps to encode mutual influences between obstacles. Ablation results further show that merging obstacle surface tokens reduces accuracy (relative $L_2$ error 0.0432), while explicit obstacle token modeling in SpiderSolver achieves superior performance (relative $L_2$ error 0.0376).

**Results on elasticity and plasticity datasets** As shown in Table 15 in Appendix F, SpiderSolver consistently outperforms all baselines, achieving the lowest relative $L_2$ error on both datasets.

**Generalization to Reynolds number and airfoil angle variations on AirfRANS.** We assess the generalizability of SpiderSolver in two scenarios using AirfRANS [35]. (1) Reynolds extrapolation (Reynolds OOD): the training set includes samples with Reynolds between 3 and 5 million, while the Reynolds of the test set spans 2 to 3 and 5 to 6 million. (2) Angle of attack extrapolation (Angles OOD): the training set covers angles from $-2.5°$ to $12.5°$, and the test set includes angles from $-5°$ to $-2.5°$ and $12.5°$ to $15°$. Table 3 shows the metrics for the OOD experiments on AirfRANS dataset. The results demonstrate that our proposed SpiderSolver achieves consistently better out-of-distribution generalization on the AirfRANS dataset.

**Generalization to shape variations of cars on ShapeNet-Car.** We compute the average point-wise Euclidean distance from each car shape to the template shape, and then select the 200 nearest shapes for training and the 100 farthest shapes for testing. The results are reported in Table 4. SpiderSolver achieves good out-of-distribution generalization to shape variations of cars on ShapeNet-Car. In Appendix D, we analyze the diversity of car geometries, and we further visualize how prediction error varies with different distances to the template shape, but observe no clear correlation between model accuracy and shape distance to template shape.

Table 4: Generalization to shape variations of cars on Shape-Net Car dataset.

| Methods | Shape-Net Car | | | |
| --- | --- | --- | --- | --- |
| | Vol $\downarrow$ | Surf $\downarrow$ | $C_D \downarrow$ | $\rho_D \uparrow$ |
| Transolver [14] | 0.0660 | 0.191 | 0.0735 | 0.9142 |
| **SpiderSolver** | **0.0510** | **0.161** | **0.0550** | **0.9222** |

## 4.2 Ablation Study and Model Analysis

**Effects of key components of the network.** We conducted the ablation study on the key components in Table 5. Referring to Equation (3), "w/o Coarse-A" denotes our SpiderSolver without the Coarse-AT and the T2P operator, and "w/o Fine-AT" denotes SpiderSolver without Fine-AT operator. "T2P-local" refers to T2P operator in Equation (5) that is computed locally in each spiderweb token. Specifically, in "T2P-local", Equation (5) is computed locally for each spiderweb token to generate the updated feature for the points belonging to the token. As shown in Table 5, removing Coarse-AT and Fine-AT considerably degrades the model's performance. Compared with removing Fine-AT, the performance drop when Coarse-AT is removed is more remarkable, highlighting its crucial role. Table 12 in Appendix E compares the runtime of SpiderSolver with and without Fine-AT.

Table 5: Ablation study of key components of SpiderSolver.

| Ablation methods | Shape-Net Car | | | | AirfRANS | | | | Blood Flow |
| --- | --- | --- | --- | --- | --- | --- | --- | --- | --- |
| | Vol $\downarrow$ | Surf $\downarrow$ | $C_D \downarrow$ | $\rho_D \uparrow$ | Vol $\downarrow$ | Surf $\downarrow$ | $C_L \downarrow$ | $\rho_L \uparrow$ | Velo $\downarrow$ |
| T2P-local | 0.0218 | 0.0741 | 0.0114 | 0.9881 | 0.0032 | 0.0055 | 0.0717 | **0.9989** | 0.0345 |
| w/o Fine-AT | 0.0236 | 0.0896 | 0.0136 | 0.9891 | 0.0022 | 0.0099 | 0.1013 | 0.9986 | 0.0347 |
| w/o Coarse-AT | 0.0324 | 0.0962 | 0.0204 | 0.9765 | 0.1258 | 0.0101 | 0.1576 | 0.9962 | 0.0634 |
| w/o Coarse-AT and Fine-AT | 0.0645 | 0.2390 | 0.0629 | 0.8585 | 0.1917 | 0.5314 | 0.1793 | 0.9915 | 0.0874 |
| **SpiderSolver** | **0.0214** | **0.0725** | **0.0096** | **0.9957** | **0.0016** | **0.0039** | **0.0665** | 0.9984 | **0.0322** |

**Ablation study on reference shape selection.** The reference shape serves only to provide a consistent indexing, and it does not affect the geometry of the resulting template shape. As shown in Table 6, our model exhibits strong robustness to the choice of reference shape on Shape-Net Car, consistently outperforming Transolver across different reference settings. Furthermore, spectral clustering results across five reference shapes demonstrate high consistency, with Dice of 0.9975 and IoU of 0.9951. Dice and IoU are computed based on the intersection and union of the five clusterings.

**Analysis of hyper-parameters on validation set.** We compare the performance and number of parameters of models with varying feature channels $C$ and network layers $L$ in Figure 6, and the varying $m_I$ and $m_B$ in Table 8. Ours consistently outperforms Transolver across different $C$ and $L$.

**Robustness to noisy boundary points.** To evaluate the robustness of spiderweb tokenization to noisy boundaries, we introduced Gaussian perturbations with varying noise levels to the car surface point cloud. As shown in Table 7, the spectral clustering and SDF-based partitioning preserve domain structure under perturbations. Furthermore, model evaluation on Shape-Net Car confirms that the robustness of patch partitioning directly contributes to stable performance against noisy boundary conditions.

Table 6: Ablation study on selection of the reference shape for Shape-Net Car.

| Shapes | Vol ↓ | Surf ↓ | $C_D$ ↓ | $\rho_D$ ↑ |
|---|---|---|---|---|
| Reference shape 1 | 0.0215 | 0.0730 | 0.0095 | 0.9934 |
| Reference shape 2 | 0.0212 | 0.0721 | 0.0103 | 0.9935 |
| Reference shape 3 | 0.0215 | 0.0725 | 0.0105 | 0.9921 |
| Reference shape 4 | 0.0210 | 0.0727 | 0.0103 | 0.9934 |
| Shape used in Table 1 | 0.0210 | 0.0738 | 0.0100 | 0.9928 |
| Mean | 0.0212 | 0.0728 | 0.0101 | 0.9930 |
| Standard Deviation | 0.0002 | 0.0006 | 0.0003 | 0.0005 |

Table 7: Robustness of spiderweb tokenization and model performance under noisy boundary perturbations on Shape-Net Car ($\sigma$ is the noise standard deviation).

| $\sigma$ | Dice ↑ | IoU ↑ | Vol ↓ | Surf ↓ | $C_D$ ↓ | $\rho_D$ ↑ |
|---|---|---|---|---|---|---|
| 0.0001 | 1.0000 | 1.000 | 0.0210 | 0.0738 | 0.0100 | 0.9928 |
| 0.001 | 0.9994 | 0.9987 | 0.0217 | 0.0739 | 0.0103 | 0.9924 |
| 0.01 | 0.9868 | 0.9743 | 0.0230 | 0.0790 | 0.0119 | 0.9919 |
| 0.03 | 0.9534 | 0.9135 | 0.0310 | 0.0829 | 0.0135 | 0.9901 |

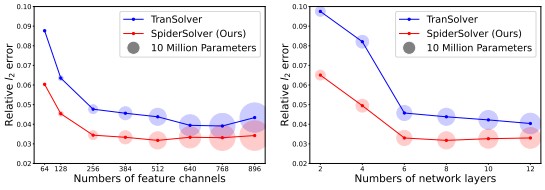

Figure 6: Performance of our model and Transolver on the Blood Flow dataset with varying channels $C$ and layers $L$ (point size $\propto$ model size).

Table 8: Ablation study of SpiderSolver with varying $m_I$ and $m_B$ on the validation set.

| $(m_I, m_B)$ | AirfRANS | | | | Blood Flow |
|---|---|---|---|---|---|
| | Vol ↓ | Surf ↓ | $C_L$ ↓ | $\rho_L$ ↑ | Velo ↓ |
| (2, 2) | 0.0026 | 0.0130 | 0.0885 | 0.9987 | 0.0372 |
| (4, 2) | 0.0021 | 0.0101 | 0.0818 | 0.9988 | 0.0366 |
| (2, 4) | 0.0019 | 0.0064 | 0.0751 | 0.9989 | 0.0359 |
| (6, 2) | 0.0020 | 0.0079 | 0.0624 | 0.9990 | 0.0352 |
| (2, 6) | 0.0019 | 0.0075 | 0.0583 | 0.9988 | 0.0350 |

**Computational cost.** On the Shape-Net Car dataset, SpiderSolver has 4.05M parameters and the network inference time for one PDE is 0.058s, compared with the 3.86M and 0.022s respectively by previous state-of-the-art method Transolver [14]. Note that it takes around 50 minutes [33] required by the traditional $k$-epsilon turbulence simulations. See Appendix E for details on inference time.

## 5 Conclusion, Limitations and Future Work

**Conclusion and impact statement.** This paper proposed a novel transformer SpiderSolver, to solve PDEs with complex domain geometry. SpiderSolver is based on the proposed spiderweb tokenization and integrates coarse-grained and fine-grained attention. Experiments on five datasets demonstrate its superiority. This work has made a fundamental contribution to the transformer architecture design in scientific computation, and may have an impact on the fast computation of numeric solutions to PDEs with complex boundaries, in the applications of science and engineering problems.

**Limitations and future work.** SpiderSolver depends on the boundary and is currently not applicable to PDEs with open-boundary domains. Moreover, it is currently applied to PDE with boundary shapes of one type of objects. In future work, we plan to extend the framework to multiple classes of shape boundaries, possibly by making the network parameters aware to different geometric classes. We also plan to extend it to be equivariant to the geometry transform of PDE boundary and domain.

## 6 Acknowledgement

This work was supported by NSFC 12125104, 12426313, 12326615, Key-Area Research and Development Program of Guangdong Province (2022B0303020003)

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

# A  Implementation Details of Spectral Clustering

The process of spectral clustering [39] can be summarized in three main stages (clusters = $k$):

**Step 1: Constructing the Affinity Matrix.** Given the input data points $(x_1, x_2, x_3, \ldots, x_n)$, the algorithm begins by treating all points as nodes in a graph. The similarity between these nodes is then quantified and stored in an affinity matrix $W$ by K-Nearest Neighbors (KNN) method. The process is as follows:

$$w_{ij} = w_{ji} = \begin{cases} \exp\left(-\dfrac{\|x_i - x_j\|^2}{2\sigma^2}\right), & \text{if } x_i \in \text{KNN}(x_j) \text{ or } x_j \in \text{KNN}(x_i) \\ 0, & \text{otherwise} \end{cases}$$

**Step 2: Constructing the Laplacian Matrix.** We use the normalized Laplacian matrix, defined as follows:
$$L = I - D^{-1/2} W D^{-1/2}$$
where $I$ is the identity matrix, $W$ is the affinity matrix, and $D = \text{diag}(d_1, ..., d_n)$, $d_i = \sum_{j=1}^{n} w_{i,j}$.

**Step 3: Computation of Eigenvectors and Clustering Completion.** First, compute the first $k$ eigenvectors $(\mathbf{u}_1, \cdots, \mathbf{u}_k)$ of the normalized Laplacian matrix $L$, forming the eigenvector matrix $U \in \mathbb{R}^{n \times k}$. Second, denote $U = (\mathbf{y}_1, \cdots, \mathbf{y}_n)^\top$, where each $\mathbf{y}_i \in \mathbb{R}^k$ corresponds to the $i$ th row of $U$. Third, cluster the points $\mathbf{y}_i$ $(i = 1, \cdots, n)$ in $\mathbb{R}^k$ with the k-means algorithm into clusters $C_1, \cdots, C_k$.

We employ standard SpectralClustering function from the scikit-learn library, configured as follows: SpectralClustering(n_clusters=clusters, affinity='nearest_neighbors', random_state=42).

# B  Visualization of Template Shapes, Boundary Clustering and Spiderweb Tokenization

In this section, we present visualizations of the template shapes of Shape-Net Car and AirfRANS, together with examples of surface clustering and spiderweb tokenization.

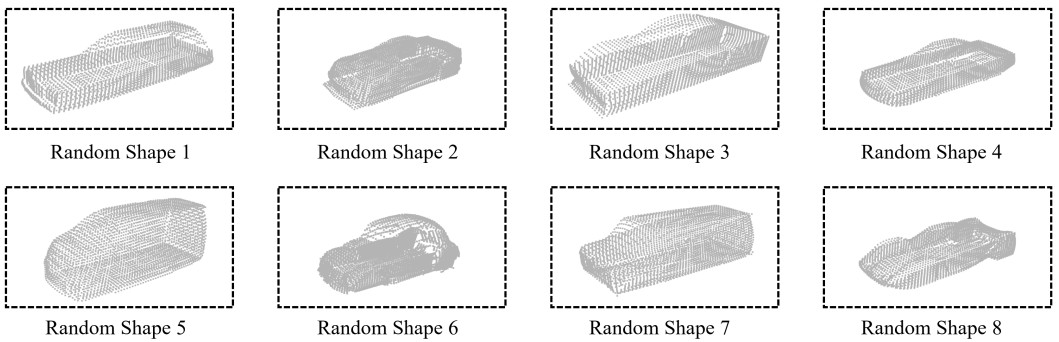

Figure 7: Eight randomly selected shapes from the Shape-Net Car dataset.

Figures 7 and 8 show the eight randomly selected shapes from the Shape-Net Car and AirfRANS, respectively. Figure 9 shows the template shapes of the Shape-Net Car and AirfRANS. Figure 10 shows the clustering of instances of cars in the Shape-Net Car, with individual clustering (i.e., separately clustering each car surface) and our aligned clustering method. The results demonstrated that our proposed aligned clustering can cluster different instances of cars with aligned clusters including the cluster indexes highlighted by different colors.

Figure 11 shows the point sets of different spiderweb tokens for an instance of a car in the Shape-Net Car dataset. Each row of Figure 11 shows the tokens corresponding to the same cluster of car surface, with increasing SDF values to the domain boundary, i.e., the car surface, from left to right. We can observe that the different tokens contain point sets with geometric structures resembling the shape of segments of a car surface.

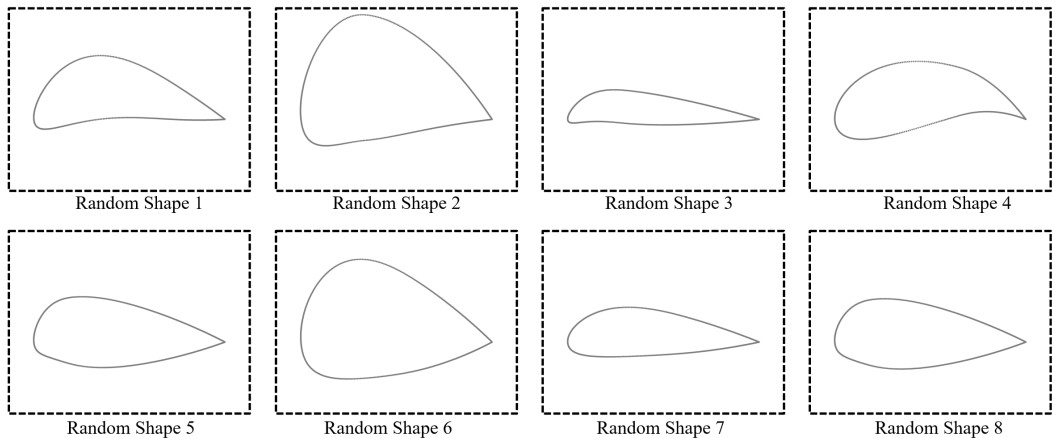

Figure 8: Eight randomly selected shapes from the AirfRANS dataset.

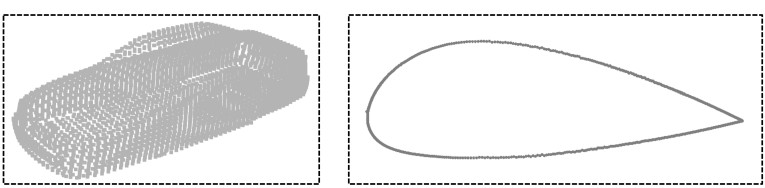

Figure 9: Template shapes of the Shape-Net Car dataset (left) and AirfRANS dataset (right).

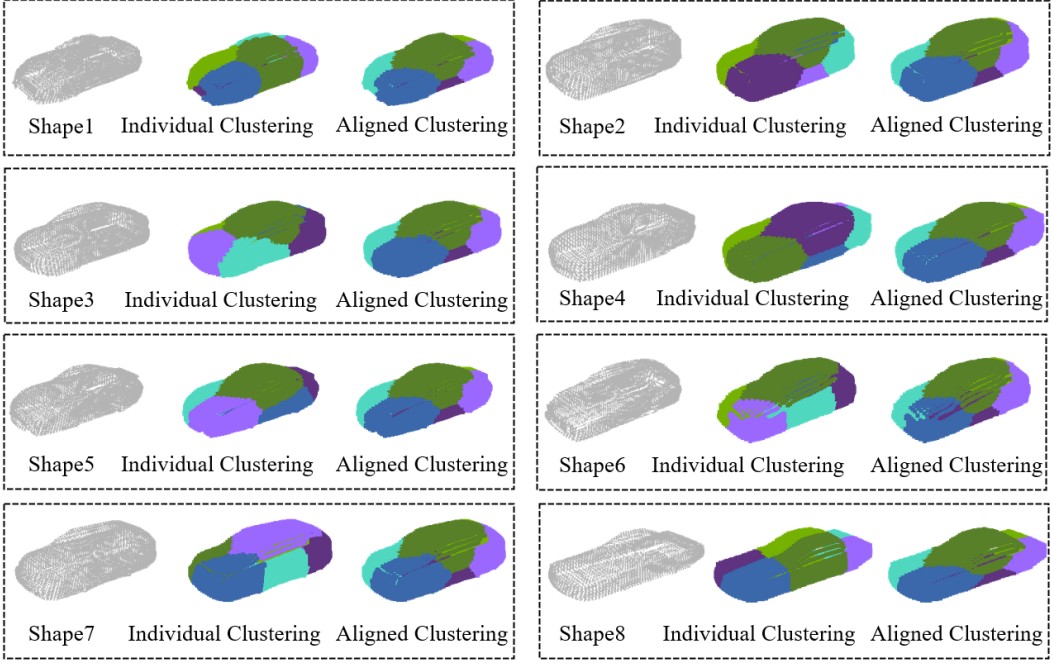

Figure 10: Clustering of car surface (i.e., the PDE domain boundary) by conducting spectral clustering on separate cars (denoted as individual clustering), and our proposed aligned clustering.

Figures 12 and 14 show an example of spiderweb tokenization of an instance of the AirfRANS and the Bounded Navier-Stokes dataset, respectively. Different colors highlight different tokens, which

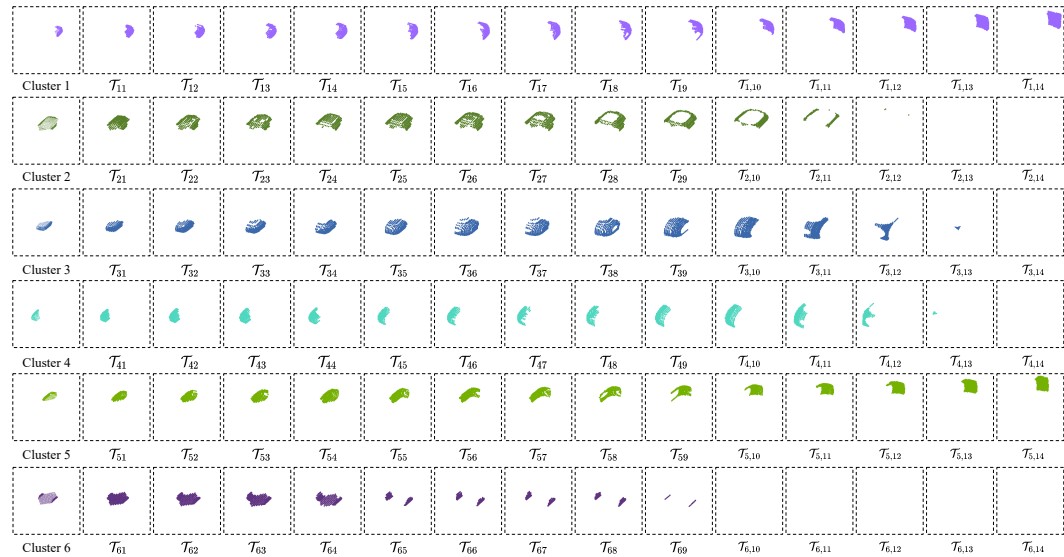

Figure 11: Point sets of different spiderweb tokens on an instance of car. The $p$-th row visualizes the point sets belonging to tokens $\mathcal{T}_{p,q}, q = 1, \cdots, m_q$ for the $p$-th ($p = 1, \cdots, m_p$) cluster of the car surface.

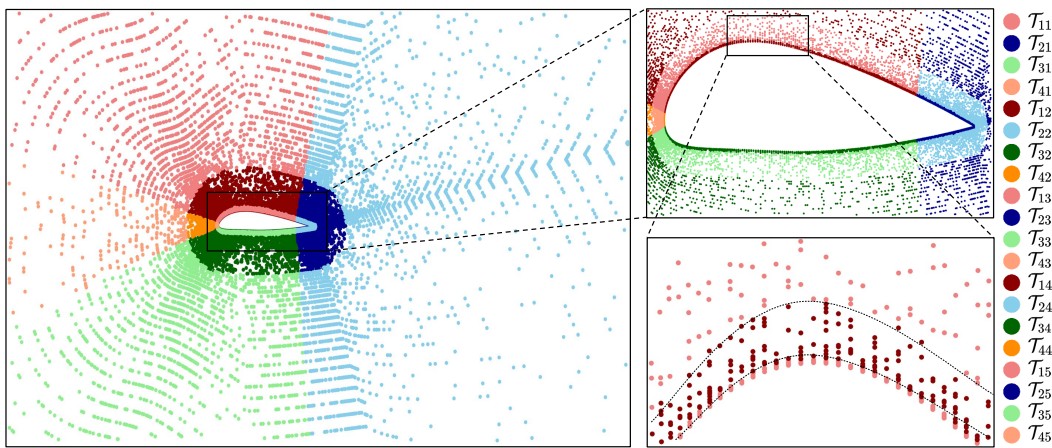

Figure 12: An example of spiderweb tokenization for an airfoil with different tokens highlighted by different colors.

well split the point sets surrounding the airfoil into sub-regions. As presented in Section 3.1.2, the determination of width, i.e., $[d_{q-1}, d_q]$ of each token, is to ensure that the regions with $[d_{q-1}, d_q]$ for $q = 1, \cdots, m_q$ contain the same number of points, thus adaptive to the different density of point clouds. For comparison, Figure 13 shows representative patches learned by Transolver on the AirfRANS dataset.

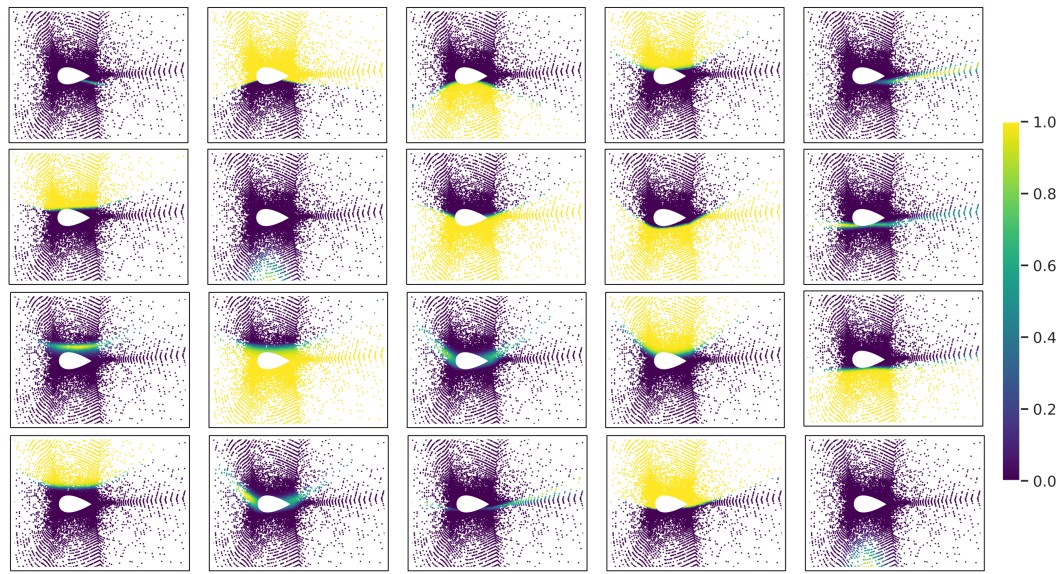

Figure 13: Representative patches learned by Transolver on the AirfRANS dataset. Colors indicate the magnitude of the weights of patches.

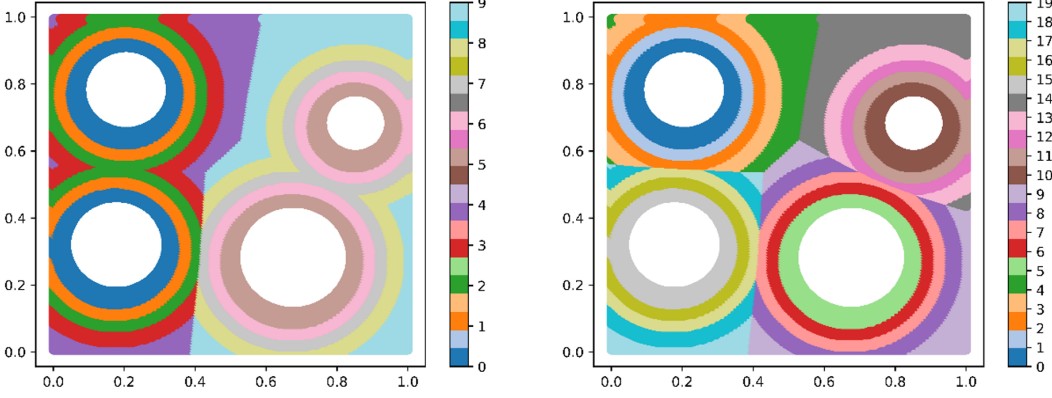

Figure 14: Spiderweb tokenization for Bounded Navier-Stokes dataset highlighted by different colors. Left: $m_B = 2, m_I = 5$; Right: $m_B = 4, m_I = 5$.

## C   Detailed Descriptions of Five Datasets and More Implementation Details of Experiments

**Shape-Net Car dataset.** This dataset is designed to simulate the behavior of a car traveling at a speed of 72 km/h, using realistic physical parameters and applying the Reynolds-averaged Navier-Stokes (RANS) equations for airflow modeling. The input models are derived from the "car" category in ShapeNet [34], with modifications to remove elements like side mirrors, spoilers, and tires, focusing more on the vehicle's aerodynamic properties. High-resolution spatial grids are used to solve the Navier-Stokes equations, simulating airflow for a 10-second duration. The time-averaged velocity and pressure fields are obtained by averaging the results from the last 4 seconds of the simulation. Each simulation involves degrees of freedom ranging from 600k to 700k, and each run takes around 50 minutes to complete. A finite element Navier-Stokes solver resolves the fluid dynamics, using the k-epsilon turbulence model and SUPG stabilization technique [40]. This setup accurately represents the flow around the vehicle, capturing complex phenomena such as boundary layers and vortex shedding at the surface.

For the Shape-Net Car dataset, the space is discretized into 32186 mesh points. Overall, a total of 889 samples with different car shapes are generated to simulate. 789 samples are used for training, and the other 100 samples are for testing, which follows that of the Transolver [14]. The model is trained to predict the velocity and pressure value for each point, then the drag coefficient can be calculated based on these estimated physical fields.

For the Shape-Net Car dataset, the input dimension of our model is $32186 \times 7$, where $32186$ represents the number of discretized points in the PDE domain, and 7 corresponds to the point coordinates, SDF values, normal vectors of boundary points and direction vectors of interior points. The output of our model is $32186 \times 4$, where $32186$ represents the number of discretized points in the PDE domain, and $4$ corresponds to the wind field of interior points and the pressure of boundary points (car surface). The drag coefficient can be calculated based on these predicted physics quantities.

**AirfRANS dataset.** AirfRANS [35] contains high-fidelity simulation data for RANS equations on airfoils that are the 4 and 5-digit series of the National Advisory Committee for Aeronautics. Each case is discretized into 32,000 mesh points. By changing the airfoil shape, Reynolds number, and angle of attack, AirfRANS provides 1000 samples, where 720 samples are used for training, 80 samples are used for validation, and 200 for the test set. The model is trained to predict the air velocity, pressure, and viscosity for each point, and calculate the lift coefficient. AirfRANS includes high-fidelity simulation data for RANS equations applied to both the NACA 4-digit and 5-digit series airfoils. Specifically, 1000 simulations were run, each defined by the airfoil, Reynolds number, and angle of attack. The goal of the simulations is to reflect realistic conditions, with the Mach number capped at 0.3 and the Reynolds number ranging from 2 million to approximately 6 million. The angle of attack varies from -5° to 15°. The simulations were performed using the steady-state RANS solver $simpleFOAM$ with the SIMPLEC algorithm [41], combined with the $k$-$\omega$ SST turbulence model [42], until the lift and drag coefficients converged.

For the AirfRANS dataset, the input dimension of our model is $32000 \times 7$, where 7 corresponds to the point coordinates, inlet velocity, SDF values, normal vectors of boundary points, and direction vectors of interior points. The output of our model is in size of $32000 \times 4$, where $32000$ represents the number of discretized points in the PDE domain, and $4$ corresponds to the wind field of interior points, the pressure of boundary points (airfoil surface), and turbulent viscosity of interior points. We aim to estimate the lift coefficient of airfoils, as well as the pressure on both interior and boundary points.

**Blood Flow dataset.** Blood Flow [36] contains the simulation data of the Navier-Stokes equations for the blood flow dynamics in the human thoracic aorta. In this dataset, the computational domains are the same, i.e., the human thoracic aorta, but the pressure and velocity at the inlet/outlets are given 500 different samples as boundary conditions. The spatial domain is represented by a tetrahedral mesh with 1656 nodes and the temporal domain is discrete with 121 temporal time points. 400 samples are used as training data, 50 samples are used as validation data and the remaining 50 as test data. The model is trained to output the velocity field of the blood flow. This dataset contains simulation data for the Navier-Stokes equations modeling blood flow dynamics in the human thoracic aorta. The aorta has one inlet (ascending aorta) and five outlets (descending aorta, left/right subclavian arteries, and left/right common carotid arteries), with velocity boundary conditions at the inlet and pressure boundary conditions at the outlets. The dataset describes the time-varying velocity and pressure over one cardiac cycle (1.2 s). Blood is modeled as a homogeneous Newtonian fluid with a density of $1060 \, \text{kg/m}^3$ and a viscosity of $0.0035 \, \text{N·s/m}^2$, assuming laminar flow in the aorta and rigid vessel walls with no-slip conditions. A total of 500 velocity/pressure curves are used as inputs, and the velocity field is simulated as the output using COMSOL Multiphysics [36].

For the Blood Flow dataset, we aim to predict the velocity field at the interior points of the aorta using pressure values at the inlets and outlets of the aorta. The geometry of the aorta remains consistent across samples, eliminating the need for aligned boundary clustering for different instances of the object surface. For the Blood Flow dataset, the spatial domain is represented by a tetrahedral mesh with 1,656 nodes, and the velocity field is characterized by a dimension of $1656 \times 1 \times 3$. Pressure values at the two inlets and four outlets are recorded across 121 time steps, with a variable dimension of $1 \times 121 \times 6$. The first two dimensions of the above two variables are replicated to match each other, and they are concatenated along the third dimension, yielding a variable dimension of $1656 \times 121 \times 9$. This is then resized to $1656 \times 1089$ and used as input to our model. The output of the model has a

dimension of $1656 \times 363$, and it is resized to $1656 \times 121 \times 3$, representing the predicted velocity field at each point across the 121 time steps.

**Bounded Navier-Stokes dataset.** The Bounded Navier-Stokes dataset simulates incompressible dye flow governed by the Navier-Stokes equations, with a Reynolds number of 256. At this regime, flow instability leads to periodic vortex shedding and the formation of a *Kármán vortex street* [43] behind cylindrical obstacles. The presence of multiple cylinders and downstream obstructions induces complex interactions that challenge predictive models. To generate the data, [37] run simulations over $10^5$ steps with fixed cylinder positions but varying initial conditions. From each trajectory, multiple frames are extracted and randomly partitioned into training and test sets. The numerical solver employs a finite difference scheme (MAC method) with CIP-based advection, enabling high-resolution flow capture while minimizing numerical dissipation.

The model predicts the physical field at the 10th time step based on the initial conditions. A total of 1,000 training sequences, 200 validation sequences and 200 test sequences are used at a spatial resolution of $64 \times 64$. Despite the presence of separate boundaries, the signed distance function (SDF) at each point is uniquely defined as the minimum distance to all boundary components. Consequently, our SDF-based spiderweb tokenization produces non-overlapping partitions, even in such complex domains.

**Darcy Flow dataset.** The Darcy Flow equation is a second-order elliptic PDE, which is defined as

$$
\begin{aligned}
-\nabla \cdot (a(x)\nabla u(x)) &= f(x), & x \in (0,1)^2, \\
u(x) &= 0, & x \in \partial(0,1)^2,
\end{aligned}
\tag{6}
$$

where $f(x)$ is the forcing function, $a(x)$ is the diffusion coefficient, and $u(x)$ is the density of fluid. We use the same parameter settings for the Darcy Flow equation as in FNO [3]. The diffusion coefficient $a(x)$ is generated from $\Psi_\sharp \mathcal{N}(0, (-\Delta + 9I)^{-2})$ with zero Neumann boundary conditions on the Laplacian operator $\Delta$. The $\Psi_\sharp$ is the point-wise push-forward operator that takes the value of 12 if $x > 0$, and 3 elsewhere. We learn the mapping from $a(x)$ to the steady state $u(x)$, fixing the forcing term $f(x) = 1$. We use $1000, 200$, and $200$ pairs of $a(x)$ and $u(x)$ in the train, validation, and test sets, respectively. The Darcy Flow equation dataset is set to four resolutions, i.e., $s = 85$, where $s$ represents that both the $a(x)$ and $u(x)$ are discretized into $s \times s$ grid.

For the Darcy Flow Dataset, the coefficient $a(x)$ results in disconnected multi-regions. In such a dataset, we can take the spatial outer rectangle boundary as the boundary to compute clustering and SDF for tokenization.

Table 9 shows more implementation details for the network structure and network training.

Table 9: Settings of hyper-parameters in the SpiderSolver network and SpiderSolver training.

| Dataset | $m_I$ | $m_B$ | $L$ | $h$ | $C$ | $d_k$ | Loss | Epochs | LR | Optimizer | Batch Size |
|---|---|---|---|---|---|---|---|---|---|---|---|
| Shape-Net Car | 5 | 4 | 8 | 8 | 256 | 32 | Relative $L_2$ | 200 | 0.001 | Adam | 1 |
| AirfRANS | 5 | 4 | 8 | 8 | 256 | 32 | Relative $L_2$ | 400 | 0.001 | Adam | 1 |
| Blood Flow | 5 | 6 | 8 | 8 | 512 | 64 | Relative $L_2$ | 500 | 0.001 | Adam | 10 |
| Bounded Navier-Stokes | 4 | 4 | 8 | 8 | 256 | 32 | Relative $L_2$ | 200 | 0.0005 | Adam | 10 |
| Darcy Flow | 8 | 8 | 8 | 8 | 256 | 32 | Relative $L_2$ | 1000 | 0.001 | Adam | 4 |

**Elasticity dataset.** The Elasticity dataset consists of simulations on unit cells with arbitrary voids at the center, where the bottom edge is clamped and tension is applied on the top. Each case takes as input a tensor of shape $972 \times 2$ representing the 2D positions of discretized points, and outputs the corresponding stress field in the shape of $972 \times 1$. A total of 1000 training samples and 200 test samples are provided, obtained using a finite element solver with about 100 quadratic quadrilateral elements. The inputs are represented as point clouds of size around 1000, and the outputs correspond to the stress fields. This dataset features an unstructured input format, which is designed to evaluate the capability of models to handle irregular geometries.

**Plasticity dataset.** The Plasticity dataset is constructed from plastic forging problems, where a block of material is impacted by a rigid die with random shapes sampled from spline interpolations. The simulations are performed on a $101 \times 31$ structured mesh with 20 time steps, resulting in 900 training samples and 80 test samples. Each case takes as input the die shape discretized into a structured mesh

of size $101 \times 31$, and outputs the deformation of each mesh point over 20 time steps in a tensor of shape $20 \times 101 \times 31 \times 4$, containing the deformation in four directions. The outputs correspond to the deformation fields induced by time-dependent die motions. This dataset emphasizes temporal dynamics and structured mesh inputs, providing a challenging benchmark for models that combine spatial and temporal learning.

## D  Shape Distribution of Shape-Net Car and Error Analysis across Shape Distance

In this section, we quantify shape variation using the distance between each car and the template shape, and analyze the distribution of car geometries. We further visualize how prediction error varies with shape differences, but observe no clear correlation between shape distance and model accuracy.

**Distance from each car shape to the template shape.** We align each shape to the template shape and compute the average point-wise Euclidean distance from all 789 training car surfaces to the template shape, as summarized in Table 10.

Table 10: Distance to template shape

| Cases | Distance (meter) |
|---|---|
| Min | 0.03254 |
| Max | 0.6296 |
| Mean | 0.1802 |
| Standard Deviation | 0.09741 |
| Top 200 closest distances | [0.03254, 0.1031] |
| Top 100 farthest distances | [0.3005, 0.6296] |

The template shape serves as an intermediate reference for aligning and clustering different car geometries. Figure 15 presents the template shape, along with the four nearest and four farthest shapes to the template shape, respectively. The distance to the template is computed as the average point-wise Euclidean distance following optimal transport (OT) alignment. As shown in Figure 1, this distance metric captures shape variation across the Shape-Net Car dataset.

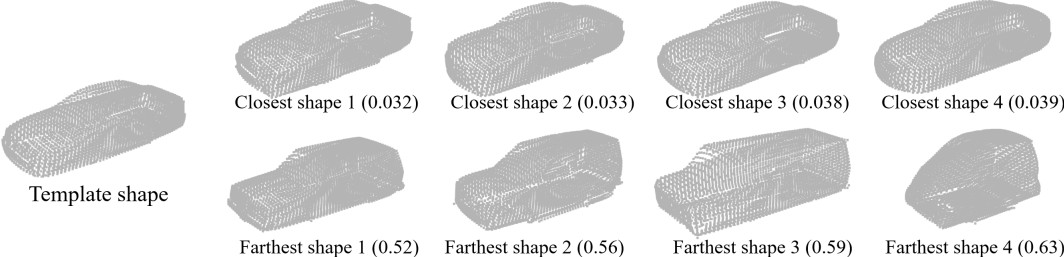

Figure 15: Template shape and closest and farthest four shapes to the template shape, with distances (meters) in parentheses.

**Shape distribution of cars on Shape-Net Car dataset.** To characterize the shape distribution, we visualize the distances from training shapes to the template shape using a histogram and a Q-Q plot in Figure 16. The histogram reveals a right-skewed distribution with a gradually decaying tail, while the Q-Q plot indicates significant deviation from normality in the upper quantiles, suggesting heavier-than-Gaussian tails. Note: A Q-Q plot assesses whether a dataset follows a reference distribution by comparing their quantiles. In the context of heavy-tailed detection, a pronounced upward deviation of the upper quantiles from the reference line indicates that the data exhibits heavier tails than the reference (e.g., normal) distribution.

**Analyzing error across shape distance of cars.** We further analyze how prediction error varies with the distance to the template shape on the test set, as shown in Figure 17 (ours) and Figure 18 (Transolver). For the prediction of our model, the two shapes farthest from the template ( 0.5 m) do

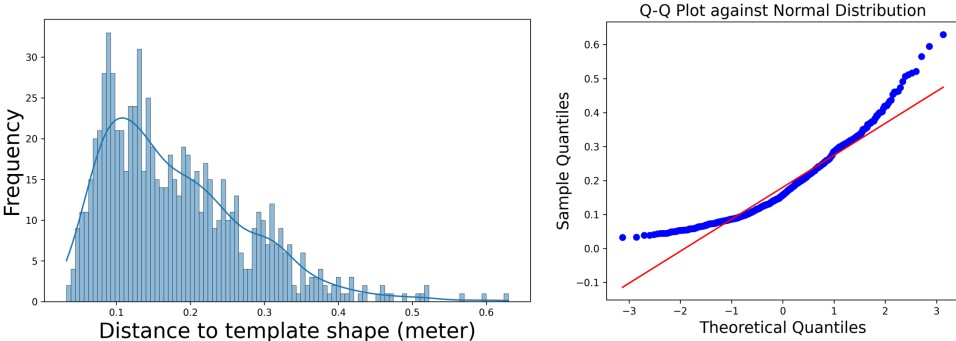

Figure 16: The histogram and Q-Q plot of different distances to template shape of Shape-Net Car Dataset.

not yield the highest prediction errors. No clear correlation is observed between shape distance and prediction accuracy.

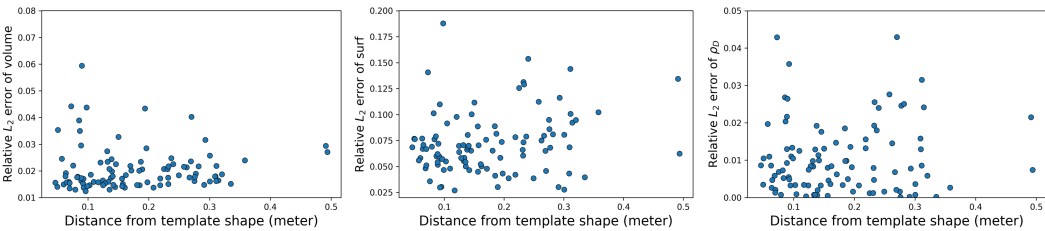

Figure 17: Our model's testing error with different car distance to template shape on Shape-Net Car Dataset.

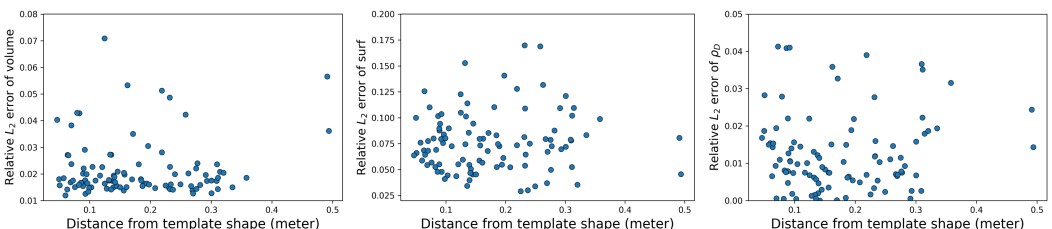

Figure 18: Transolver's testing error with different car distance to template shape on Shape-Net Car Dataset.

# E  Inference Time and Evaluation Metrics

**Inference time.** The inference time of our model comprises the spiderweb tokenization (includes optimal transport-based matching) and the network forward pass. Spectral clustering is only performed once for each shape during training; during inference, cluster labels are propagated from the template shape to the test shape via optimal transport-based boundary points correspondences. For the Shape-Net Car dataset, spectral clustering is performed on 3,586 surface points using only the top $m_B$ eigenvectors of the Laplacian matrix (e.g., 6 eigenvectors for $m_B = 6$), which keeps the computational cost manageable. We report the model inference time in Table 11. Note that the prediction commonly takes around 50 minutes, which is required by the traditional $k$-epsilon turbulence simulations.

**Comparison of models with and without Fine-AT.** We also compare the inference time of the model with and without the Fine-AT on the Shape-Net Car dataset, as shown in Table 12.

Table 11: Inference time and number of parameters on the Shape-Net Car dataset.

| Different terms | SpiderSolver (Ours) | Transolver |
|---|---|---|
| Time of spectral clustering (only model training) | 0.26s | N/A |
| Number of surface points of car | 3586 | 3586 |
| Time of point matching via optimal transport | 0.012s | N/A |
| Time of interior domain partition before model forward | 0.0032s | N/A |
| Time of model forward | 0.043s | 0.022s |
| Total inference time | 0.058s | 0.022s |
| Number of parameters | 4050340 | 3860804 |

Table 12: Comparison of Inference time of Fine-AT on the Shape-Net Car dataset.

| Setting | Model Forward time (s) | Total Inference time (s) |
|---|---|---|
| SpiderSolver (Ours) w/o Fine-AT | 0.032 | 0.047 |
| SpiderSolver (Ours) w/ Fine-AT | 0.043 | 0.058 |

Since our evaluation is based on practical tasks of different datasets, we include several design-oriented metrics for physics fields, which will be introduced as follows.

**Relative $L_2$ error for physics fields.** The relative $L_2$ error $L(u, \hat{u})$ between the true physics field $u$ and the model's predicted field $\hat{u}$ is calculated as follows:

$$L(u, \hat{u}) = \frac{\|u - \hat{u}\|_2}{\|u\|_2}. \tag{7}$$

**Drag and lift coefficients.** For the Shape-Net Car and AirfRANS datasets, drag and lift coefficients are computed from the estimated physics fields. For unit-density fluids, the coefficients are given by:

$$C = \frac{2}{v^2 A} \left( \int_{\partial\Omega} p(\omega) \left( \hat{n}(\omega) \cdot \hat{i}(\omega) \right) d\omega + \int_{\partial\Omega} \tau(\omega) \cdot \hat{i}(\omega) d\omega \right), \tag{8}$$

where $v$ is the inlet flow velocity, $A$ is the reference area, and $\partial\Omega$ is the object surface. The pressure $p$, the outward unit normal $\hat{n}$, the flow direction $\hat{i}$, and the wall shear stress $\tau$ are defined as usual. The shear stress $\tau$ is typically much smaller than the pressure term and can be approximated by the air velocity near the surface [44]. We use the setting of drag and lift coefficient from Transolver [14].

**Spearman's rank correlation for drag and lift coefficients.** Given $K$ test samples, let the true drag or lift coefficients be $D = \{D^1, \ldots, D^K\}$ and the predicted coefficients be $\hat{D} = \{\hat{D}^1, \ldots, \hat{D}^K\}$. The Spearman rank correlation is computed by ranking both sets of coefficients and then calculating the Pearson correlation between the ranks [14]:

$$\rho = \frac{\text{cov}(R(D), R(\hat{D}))}{\sigma_{R(D)}\sigma_{R(\hat{D})}}, \tag{9}$$

where $R$ is the ranking function, cov denotes covariance of the ranks, and $\sigma$ is the standard deviation. Higher $\rho$ indicates better alignment between the predicted and true coefficients [45].

# F  Results on Bounded Navier-Stokes, Darcy Flow, elasticity and plasticity datasets

Tables 13 and 14 show the results on Bounded Navier-Stokes and Darcy Flow, respectively. Table 15 shows the results on Elasticity and Plasticity datasets.

Table 13: The results on the Bounded Navier-Stokes dataset.

| Methods | Relative $L_2$ error ↓ |
|---|---|
| FNO [3] | 0.0472 |
| GNOT [32] | 0.0589 |
| Galerkin [13] | 0.4908 |
| Factformer [11] | 0.0452 |
| Transolver [14] | 0.0555 |
| DeepLag [37] | 0.0382 |
| **SpiderSolver** | **0.0376** |

Table 14: The results on the Darcy Flow dataset.

| Methods | Relative $L_2$ error ↓ |
|---|---|
| FNO [3] | 0.0108 |
| WMT [46] | 0.0082 |
| GEO-FNO [6] | 0.0108 |
| F-FNO [16] | 0.0077 |
| Galerkin [13] | 0.0084 |
| LSM [47] | 0.0065 |
| Transolver [14] | 0.0068 |
| **SpiderSolver** | **0.0064** |

## G  More Visualization Results on Shape-Net Car and Blood Flow

Figure 19 visualizes the Shape-Net Car estimation results by Transolver and our model. The visual results demonstrate that our approach achieved less error in the estimation of physical fields, especially the press value in the car surfaces.

Figure 20 visualizes the blood flow estimation results by different compared methods. The estimation absolution error, i.e., the point-wise $L_2$ norm of the difference between ground truth and prediction values of different models, is highlighted by heatmaps. The visual results demonstrate that our approach achieved less error in the estimation of physical fields.

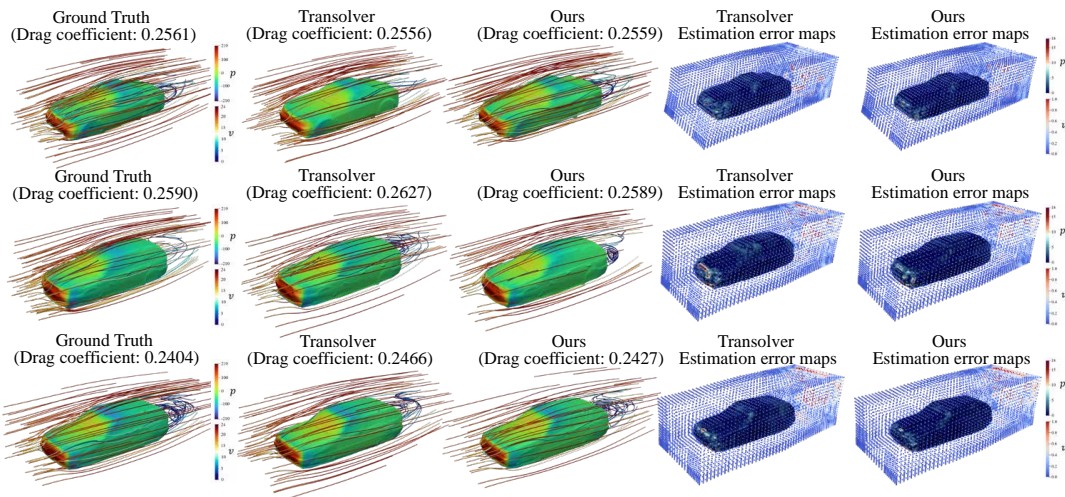

Figure 19: Comparisons of the Shape-Net Car estimation by Transolver and our model, with the estimation error maps, i.e., point-wise $L_2$ norm of the difference between ground truth and prediction.

Table 15: The results (Relative $L_2$ error) on the Elasticity and Plasticity datasets.

| Methods | Elasticity ↓ | Plasticity ↓ |
|---|---|---|
| LSM | 0.0218 | 0.0025 |
| GALERKIN | 0.0240 | 0.0120 |
| OFORMER | 0.0183 | 0.0017 |
| GNOT | 0.0086 | 0.0336 |
| ONO | 0.0118 | 0.0048 |
| Transolver | 0.0064 | 0.0012 |
| **SpiderSolver** | **0.0061** | **0.0011** |

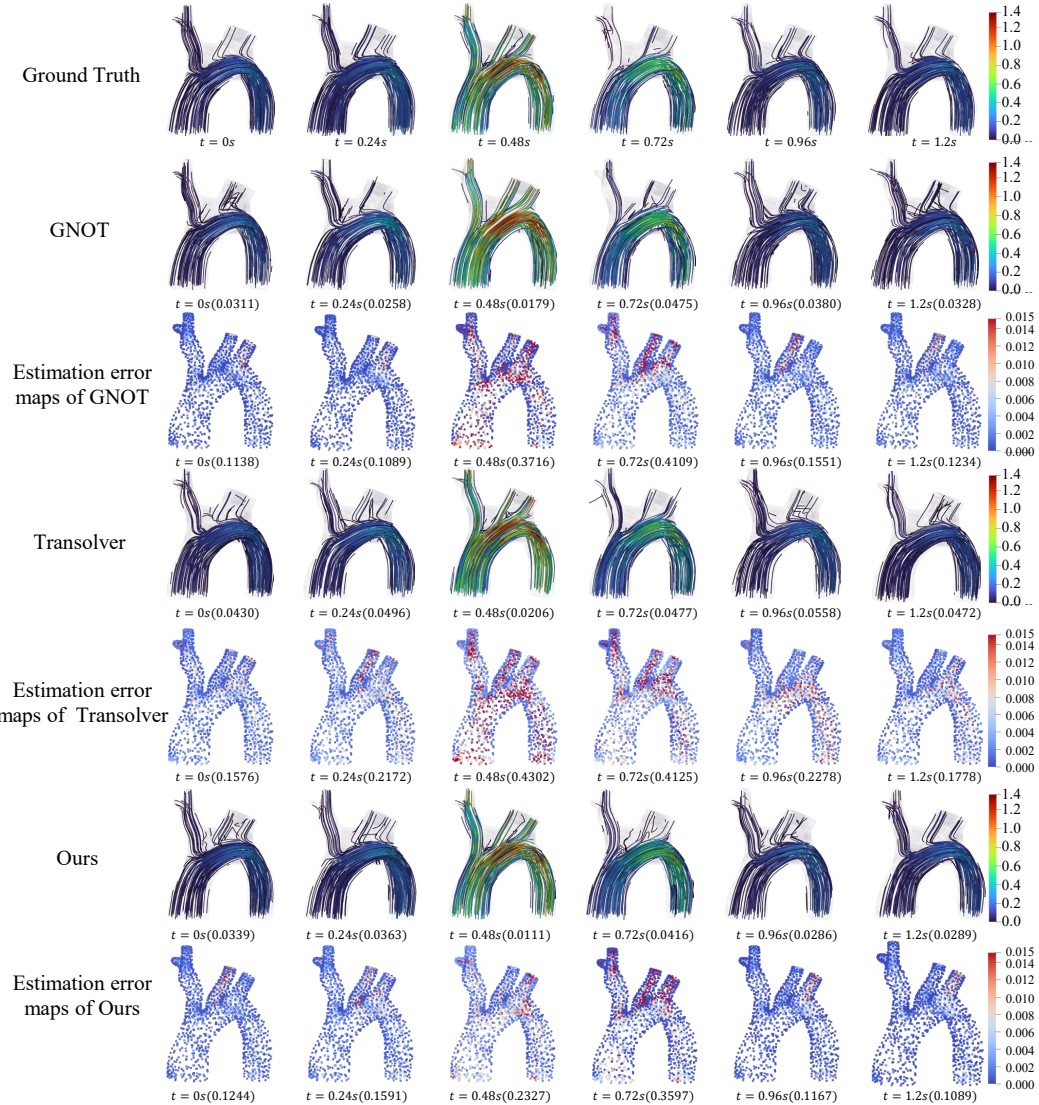

Figure 20: Comparisons of the blood flow estimation by different methods, with the estimation error maps, i.e., point-wise $L_2$ norm of the difference between ground truth and prediction.

