# OpenReview forum: "SpiderSolver: A Geometry-Aware Transformer for Solving PDEs on Complex Geometries"
_NeurIPS.cc/2025/Conference — NeurIPS 2025 poster_

### Official Review · Reviewer_1P4P · 2025-06-30

**Clarity:** 4
**Significance:** 3
**Originality:** 3
**Rating:** 5
**Confidence:** 5

**Summary:**

The paper focuses on improving learning solution operators for PDE on various geometries. First, the authors proposed a tokenization method  based on the eigenvalue of Beltrami-Laplacian operator $\Delta$ on boundary $\partial \Omega$, which characterizes the intrinsic geometry structure of $\partial \Omega$. By spectral clustering, the domain is decomposed into subregions. Second, the author proposed an architecture of neural operators that has two components. A coarse attention handles feature of the clustered points from the first step, and a fine attention handles features from original points.

Experiments include three datasets of challenging geometry shapes: Shape-Net Car, AirfRANS, Blood Flow. The proposed method outperforms all baselines.

**Questions:**

N.A.

**Ethical Concerns:**

["NO or VERY MINOR ethics concerns only"]

**Final Justification:**

I thank the author for the additional experiment in their rebuttal to respond my concerns.

It seems that the proposed method has certain advantage particularly near boundary and contact surface due to its geometric embedding. It seems that the embedding technique can be complimentary to other neural operators as well, e.g., the performance of Transolver is improved in the additional experiment.

As for the proposed architecture, a design of local-global splitting is recently getting popular, for example, [1]. The proposed method has achieved a good balance between accuracy and computational efficiency due to local (within patch) attention and coarse (patch-wise) attention.

Therefore, the overall contribution of this paper is positive. I will change my score to 5.

[1] Liu-Schiaffini, M., Berner, J., Bonev, B., Kurth, T., Azizzadenesheli, K., & Anandkumar, A. (2024). Neural operators with localized integral and differential kernels. arXiv preprint arXiv:2402.16845.

**Limitations:**

N.A.

**Paper Formatting Concerns:**

N.A.

**Quality:**

3

**Strengths And Weaknesses:**

**Strength**

The idea of tokenizing points by spectral clustering is novel and reasonable. The paper is nicely presented and easy to follow.

**Weakness**

The proposed dual-attention structure causes overhead parameters and computation (see Table 9). One alternative would be simply using clustering as additional feature. Also, such clustering feature should naturally be utilized by the "slicing" structure of Transolver. Hence, I strongly recommend the authors to conduct an experiment of combining your tokenization technique with Transolver.

---

> ### Author Rebuttal · Authors · 2025-07-31
>
> ### **Q1: Using clustering as an additional feature.**
>
> We thank the reviewer for their thoughtful comments. In response, we designed a dedicated experiment to evaluate this aspect.
> We first perform aligned boundary clustering on car surfaces following our method (Lines 153–165, page 5). The resulting spectral clustering labels are encoded into one-hot vectors, which are then passed through a two-layer MLP to match the dimensionality of Transolver’s initial features. These features are added to Transolver’s original initial features before entering the Transformer block. All other components of Transolver remain unchanged.
>
>
> Table R4-1.  Comparative results of Transolver variants incorporating our clustering with different clusters on Shape-Net Car.
>
> | Model              | Vol ↓ | Surf ↓ | $C_D$ ↓ | $ \rho_D $ ↑ |
> |-------------------|-----------|--------|--------------|----------------|
> | Transolver   |  0.0228 |   0.0793  |  0.0129   |  0.9916     |
> | SpiderSolver (Ours)  | 0.0210  | 0.0738 | 0.0100  | 0.9928   |
> | Transolver with 4 clusters   |  0.0212 | 0.0776    |  0.0126 |   0.9894     |
> | Transolver with 6 clusters   |  0.0220 |   0.0789  |  0.0123 |    0.9913   |
> | Transolver with 8 clusters   |  0.0217 |  0.0782   |   0.0119|   0.9906    |
> | Transolver with 10 clusters   | 0.0212  |   0.0778  | 0.0115  |   0.9919    |
>
> As shown in Table R4-1, integrating our clustering features into the initial features of Transolver yields a slight performance improvement. However, the gain remains limited, and the model still falls short of the performance achieved by our full SpiderSolver.
> This indicates that while clustering provides useful geometric priors, the holistic design of SpiderSolver
> (including Spiderweb tokenization, fine- and coarse-grained attention) is essential for fully leveraging boundary-aware representations.
>
> ### **Q2: Combine our tokenization with Transolver's slicing structure.**
>
> We appreciate the reviewer’s insightful suggestion and accordingly conducted a targeted experiment.
> We replace Transolver’s learnable slicing mechanism with our deterministic Spiderweb tokenization to define the patch structure. We set $m_I = 8$ and  $m_B = 4$, resulting in the same number of patches as used in Transolver.
> The remaining components of Transolver, including the architecture and training procedure, are kept unchanged.
>
> Table R4-2.  Comparative results of Transolver variants incorporating Spiderweb tokenization on Shape-Net Car.
>
> | Model              | Vol ↓ | Surf ↓ | $C_D$ ↓ | $\rho_D$ ↑ |
> |-------------------|-----------|--------|--------------|----------------|
> | Transolver   |  0.0228 |   0.0793  |  0.0129   |  0.9916     |
> | SpiderSolver (Ours)  | 0.0210  | 0.0738 | 0.0100  | 0.9928   |
> | Transolver with Spiderweb tokenization   |  0.0223 | 0.0752    | 0.0112  |    0.9920   |
>
> As shown in Table R4-2,
> replacing Transolver’s original slicing mechanism with our spiderweb tokenization improves its performance. However, it still underperforms compared to our full SpiderSolver model.
> This performance gap arises because SpiderSolver further incorporates fine-grained attention to model interactions between boundary points and nearby interior points, enabling more precise information flow across the domain.
>
> ###  **Conclusion and future work.**
>
> We thank the reviewer 1P4P for offering a valuable perspective on our model architecture. The suggestions provide promising directions for future exploration, and we plan to investigate these ideas in subsequent work.
> Beyond the reviewer’s suggestions, we have also identified several potential avenues for future research based on the overall findings and limitations of our current study.
>
>
> In this paper, we propose SpiderSolver, a transformer-based model tailored for solving PDEs over complex domain geometries.
> At the core of SpiderSolver is the novel spiderweb tokenization, which enables flexible geometric decomposition of the domain. The model integrates both coarse-grained attention (across region-level tokens) and fine-grained attention (between boundary and interior points), capturing multiscale interactions essential to PDE dynamics.
>
> We originally evaluated SpiderSolver on five representative benchmarks: ShapeNet Car, AirfRANS, Blood Flow, Bounded Navier-Stokes, and Darcy Flow.
> With the newly added experiments on the Elasticity and Plasticity datasets, our evaluation now spans seven benchmarks with diverse physics and geometries.
> Across all these settings, SpiderSolver consistently achieves state-of-the-art performance.
> These results demonstrate the generality and practical value of our approach, highlighting its potential for solving real-world scientific problems involving complex domains.
>
> In future work, we plan to extend the framework to multiple classes of shape
>  boundaries, possibly by making the network parameters aware to different geometric classes. We
>  also plan to extend it to be equivariant to the geometry transform of the PDE boundary and domain.

---

> > ### Comment · Reviewer_1P4P · 2025-08-05
> >
> > I thank the author for the additional experiment in their rebuttal to respond my concerns.
> >
> > It seems that the proposed method has certain advantage particularly near boundary and contact surface due to its geometric embedding. It seems that the embedding technique can be complimentary to other neural operators as well, e.g., the performance of Transolver is improved in the additional experiment.
> >
> > As for the proposed architecture, a design of local-global splitting is recently getting popular, for example, [1]. The proposed method has achieved a good balance between accuracy and computational efficiency due to local (within patch) attention and coarse (patch-wise) attention.
> >
> > Therefore, the overall contribution of this paper is positive. I will change my score accordingly.
> >
> > [1] Liu-Schiaffini, M., Berner, J., Bonev, B., Kurth, T., Azizzadenesheli, K., & Anandkumar, A. (2024). Neural operators with localized integral and differential kernels. arXiv preprint arXiv:2402.16845.

---

> > > ### Author Response · Authors · 2025-08-05
> > >
> > > We thank the reviewer for the constructive and encouraging feedback.
> > >
> > > We will cite the recommended reference [1] and include a discussion to place our method in the context of recent architectural advances.
> > >
> > > Thank you again for your thoughtful evaluation and positive comments on this work.
> > >
> > > [1] Liu-Schiaffini, M., Berner, J., Bonev, B., Kurth, T., Azizzadenesheli, K., & Anandkumar, A. (2024). Neural operators with localized integral and differential kernels. arXiv preprint arXiv:2402.16845.

---

### Official Review · Reviewer_hsJq · 2025-07-02

**Clarity:** 3
**Significance:** 2
**Originality:** 2
**Rating:** 4
**Confidence:** 4

**Summary:**

This paper proposes SpiderSolver, a geometry-aware transformer that introduces spiderweb tokenization for handling complex domain geometry and irregularly discretized points. This method partitions the irregular spatial domain into spiderweb-like patches, guided by the domain boundary geometry. SpiderSolver leverages a coarse-grained attention mechanism to capture global interactions across spiderweb tokens and a fine-grained attention mechanism to refine feature interactions between the domain boundary and its neighboring interior points. Experimental results demonstrate that SpiderSolver consistently achieves state-of-the-art performance across different datasets and metrics.

**Questions:**

In eq. 5, how is the initial point-wise feature $x^0$ computed? Since spiderweb token features $\hat{f}^{l}$ is shared for each patch,  the initial point-wise feature must show significant difference among points.

**Ethical Concerns:**

["NO or VERY MINOR ethics concerns only"]

**Final Justification:**

My concerns, mostly on the robustness of the proposed method, have been addressed by authors' rebuttal.

**Limitations:**

The proposed SpiderSolver is applied to PDE with boundary shapes of one type of objects. If the shape of objects vary dramatically, the tokenization transferred from the template may be unreliable. The spectral clustering for a new object with significantly new shape will be time-consuming. The attention transferred is unreliable too.

**Paper Formatting Concerns:**

No major formatting issues.

**Quality:**

3

**Strengths And Weaknesses:**

Strengths:
1. SpiderSolver is a geometry-aware Transformer-based PDE solver, which is designed to partition the domain into spiderweb-like patches utilizing physical and geometric knowledge to reduce the computation cost of attention.
2. SpiderSolver adopts the multi-grained attention mechanism to capture the intricate physical correlation of complex boundaries and interior points. The fine-grained attention can extract the interact features between points of domain boundary and their near points in the interior of domain to model the intricate  influence of boundary shapes on solutions.
3. Extensive experimental results are provided.

Weaknesses:
1. Please give a discussion on the robustness of spiderweb tokenization to the noise in boundary point coordinates.
2. The time used for spectral clustering is not provided.
3. The spiderweb tokenization generates patches that vary notably in size. Please give a discussion on the effect of variable patch size.
4. In the experiment of generalization to shape variations of cars, if the shape of cars vary significantly, the patches obtained by spiderweb tokenization will vary in space across different car instances, how does the attention between patches learned by nearest shapes transfer to patches with varying position?  Please give a discussion on this.

---

> ### Author Rebuttal · Authors · 2025-07-31
>
> ### **Q1: Robustness of spiderweb tokenization to noisy boundary points.**
> We first quantify the robustness of the spiderweb tokenization using standard similarity metrics under increasing noise levels. We then provide an analysis explaining why spectral clustering and SDF-based slicing remain structurally stable under such perturbations. Finally, we evaluate the performance of our model when exposed to noisy boundary conditions at the inference phase.
>
> ***(1) Robustness of spiderweb tokenization to boundary noise.***
> To assess the robustness of our spiderweb tokenization to noise on boundary points, we introduce Gaussian noise with zero mean and standard deviation $\sigma$ to the car surface point cloud, applied independently to each point coordinate. We then compare the resulting partitioning of the PDE domain before and after noise injection using Dice score and Intersection over Union (IoU)，as shown in Table R3-1.
>
> Table R3-1. Robustness of spiderweb tokenization under car boundary perturbation ($\sigma$ is noise standard deviation).
> | $\sigma$ | Dice ↑ | IoU ↑ |
> |--|-|-|
> | 0.0001|1.000 | 1.000 |
> | 0.001 | 0.9994 | 0.9987|
> | 0.01|0.9868 | 0.9743 |
> | 0.03 |0.9534 | 0.9135 |
>
> **Spectral clustering preserves structure under noisy boundaries.**
> The spiderweb tokenization relies on spectral clustering of the car surface. While noise perturbs the surface point cloud locally, it does not significantly alter the global geometry of the car. Spectral clustering uses the eigenvectors of the graph Laplacian to embed the data in a lower-dimensional space, which enhances global structure while suppressing local fluctuations. As a result, the derived tokenization of the PDE domain, remains stable despite surface noise.
> **SDF-based partitioning is robust to local perturbations.**
> The interior partitioning of the PDE domain is based on signed distance (SDF) values to the car surface. Since the SDF calculation inherently reflects the noisy surface, any local perturbation only causes a consistent increase or decrease in nearby SDF values. Thus, the overall structure of SDF-based slicing remains largely unaffected, further contributing to the robustness of the tokenization.
>
> ***(2) Model performance under noisy boundary conditions.***
> We evaluate the model's robustness by training on clean data and testing on inputs with noisy boundary point clouds, as shown in Table R3-2.
> The robustness of patch partitioning within the PDE domain under boundary perturbations (Table R3-1) contributes to the overall stability of the model against such noise.
>
> Table R3-2. Model performance under noisy boundary conditions on Shape-Net Car  ($\sigma$ is noise standard deviation).
>
> | $\sigma$ | Vol ↓ | Surf ↓ | $ C_D $ ↓ | $ \rho_D $ ↑ |
> |---|--|-|--|--|
> | 0.0001 |0.0210 |0.0738 | 0.0100 |0.9928 |
> | 0.001 | 0.0217 |0.0739 | 0.0103 | 0.9924 |
> | 0.01 | 0.0230| 0.0790 | 0.0119 | 0.9919 |
> | 0.03 | 0.0310|0.0829 | 0.0135 | 0.9901 |
>
> ### **Q2: The time used for spectral clustering is not provided.**
> The runtime of the spectral clustering is reported in Table 9 on page 20 of our manuscript,
> ae well as the time of point matching via optimal transport, time of interior domain partition before model forward, time of model forward.
> We will refer to Table 9 at more appropriate locations in the main text to prevent them from being overlooked.
>
> ### **Q3: Discussion on the effect of variable patch size.**
>
> We begin by explaining how the patch size variation naturally arises from our design, then discuss the physical motivations behind using non-uniform spatial resolution, and finally present an ablation study comparing against uniform partitioning.
>
> ***Origin of variable patch size in spiderweb tokenization.***
> The variation in patch size is an inherent result of our spiderweb-based tokenization scheme. This strategy allocates finer patches near the object surface (e.g., the car), while assigning coarser patches to regions farther away. Consequently, patches closer to the geometry have smaller spatial areas, reflecting higher local resolution.
>
> ***Physical rationale for non-uniform patching.***
> This non-uniform partitioning is well aligned with the nature of flow fields. Near the object, the flow exhibits rapid variations in velocity and direction. Smaller patches in these regions are better suited for capturing complex local dynamics. In contrast, regions far from the object typically resemble uniform free-stream behavior, where the physics are smoother and do not require fine spatial resolution.
>
> ***Ablation with uniform patch size.***
> To evaluate the impact of patch size variability, we conducted an ablation experiment using uniform spatial partitioning, dividing the PDE domain into equal-sized patches based solely on coordinates. The results, reported in Tables R3-3 and R3-4, demonstrate the effectiveness of our spiderweb-based variable-resolution strategy.
>
> Table R3-3. Effect of uniform patch size on model performance on Shape-Net Car.
>
> | Model | Vol ↓ | Surf ↓ | $ C_D $ ↓ | $ \rho_D $ ↑ |
> |---|---|-|---|---|
> | SpiderSolver (Ours) | 0.0210  | 0.0738 | 0.0100 | 0.9928 |
> |Transformer with uniform patch size  |  0.0271  | 0.0996 | 0.0192| 0.9699 |
>
> Table R3-4. Effect of uniform patch size on model performance on AirfRANS.
>
> | Model | Vol ↓ | Surf ↓ | $C_L$  ↓ | $\rho_L$ ↑ |
> |---|---|--|---|---|
> | SpiderSolver (Ours)  |0.0017  | 0.0043 | 0.0741  | 0.9988   |
> | Transformer with uniform patch size   |  0.0048  |  0.0195   |  0.1356  |    0.9963   |
>
> ### **Q4: How tokenization and attention transfers reliably under significant car shape variation.**
>
> We first analyze the results from generalization experiments, then explain how tokenization and attention remain reliable under significant car shape variation from a methodological perspective.
>
> ***(1) Support from generalization experiments.***
> As detailed in lines 329-339 on page 8 and Appendix C on pages 18-19,
> we compute the average point-wise Euclidean distance from each car shape to the template shape, and then select the 200 nearest shapes for training and the 100 farthest shapes for testing.
> As shown in Figure 17 (page 19), there is no discernible correlation between shape distance and prediction error across test samples. This lack of performance degradation, even for cars exhibiting substantial geometric differences, demonstrates that our tokenization and attention mechanisms generalize effectively. The learned representations maintain semantic alignment across diverse car shapes, enabling reliable transfer of information between varying geometries.
>
> ***(2) Methodological explanation of tokenization and attention transfer.***
>
> **Semantic consistency through distance-based interior partitioning:**
> In our method, the interior domain is partitioned by leveraging both the spectral clustering of the boundary and the signed distance of each interior point to the boundary surface. This joint strategy ensures that token regions reflect the geometric layout of each shape.
>
> **Alignment across varying shapes via optimal transport:**
> To account for shape variation, we apply optimal transport (OT) to align boundary points between different car instances. This alignment provides a pointwise correspondence that preserves the semantic structure across shapes. As a result, although patch positions (by our spiderweb tokenization) may shift spatially due to shape differences, the relative semantic meaning of each patch (e.g., “above the car roof” or “near the rear bumper”) remains consistent.
>
> **Robust attention modeling across aligned patches:**
> Since attention in our model is learned over these semantically aligned tokens, the learned dependencies generalize across instances even when geometric layouts vary. This semantic alignment is essential for robust token-level attention and enables effective knowledge transfer across diverse shapes.
>
> ### **Q5: Computation of the initial point-wise feature $x^{0}$，it must show significant difference among points.**
>
> The initial point-wise feature $x^0$ is obtained by passing the model's input through a two-layer MLP. Although the same MLP is applied to all points, the input itself encodes meaningful spatial and physical differences between points, ensuring that the resulting features $\mathbf{x}^l$  remain distinct across the domain.
> We summarize below the specific input representations used for different tasks in our experiments, as shown in Table R3-5.
>
> Table R3-5. Specific input representations for different physical modeling tasks.
>
> | Datasets | Input of our model |
> |---|---|
> | Shape-Net Car | Point cloud, SDF, boundary normals, interior directions |
> | AirfRANS  | Point cloud, inlet velocity, SDF, boundary normals,  interior directions |
> | Blood Flow |  Point cloud, time-varying pressure values at the aortic inlets and outlets |
> | Bounded Navier-Stokes | Regular grid,  obstacle locations, initial physical field|
> | Darcy Flow |  Regular grid,  point-wise diffusion coefficient   |
> | Elasticity  | Point cloud of elasticity material (include void) |
> | Plasticity  | Point cloud of plasticity material, time indexing, shape of rigid die |
>
> ### **Q6: The spectral clustering for a new object with significantly new shape will be time-consuming.**
>
> During the testing phase, spectral clustering is no longer required. Instead, we perform point-wise matching between the new shape and the reference shape via optimal transport (OT) to directly retrieve the corresponding clustering result.
> As shown in Table 9 on page 20, the time of point matching via optimal transport is 0.012s for the
> Shape-Net Car dataset.
> The runtime of the OT depends on the number of discrete points on the car surface and is independent of the geometric differences between shapes.
> It is important to contextualize this overhead relative to traditional numerical solvers. For example, a conventional $k$–$\varepsilon$ turbulence simulation typically takes around 50 minutes for a single prediction of the Shape-Net Car.

---

> > ### Comment · Reviewer_hsJq · 2025-08-06
> > **Official comment to authors' rebuttal**
> >
> > Thank authors for their very detailed responses and additional experiments. My concerns have been resolved by the responses. I decide to raise my score accordingly.

---

### Official Review · Reviewer_urfu · 2025-07-03

**Clarity:** 2
**Significance:** 3
**Originality:** 3
**Rating:** 5
**Confidence:** 4

**Summary:**

This paper introduces SpiderSolver, a geometry-aware Transformer designed to solve partial differential equations (PDEs) in irregular domains. SpiderSolver introduces spiderweb tokenization, which divides the domain into sections based on the shape's boundaries and distances, allowing for effective attention across different scales. When evaluated on five diverse PDE datasets with intricate geometries, SpiderSolver consistently achieves top performance and shows enhanced generalization compared to existing approaches.

**Questions:**

1. The results reported in Table 3 of the Transolver do not exactly match those in Table 5. Could you please clarify the differences in the evaluation process?

2. Instead of aligning the boundary clustering for different instances of the object, what would happen if we performed boundary clustering independently for each instance? Would this approach improve performance? This could also serve as an interesting ablation study.

**Ethical Concerns:**

["NO or VERY MINOR ethics concerns only"]

**Final Justification:**

The authors have addressed my concerns, including rewriting several sections of the paper, adding new ablations, and discussing some important limitations. I have increased the score.

**Limitations:**

yes. The proposed model requires more time. It should be mentioned in the limitations.

**Paper Formatting Concerns:**

No concerns.

**Quality:**

3

**Strengths And Weaknesses:**

## Strength
1. Tokenizing irregularly structured domains, such as graph/mesh/point-cloud, is a challenging problem. The approach introduced here is novel to my knowledge and might be useful even outside of the problem of PDE solving.

2. The proposed technique achieves better empirical results compared to the baselines.

## Weaknesses

1. Although the paper is mostly well-written and clear, the paragraph on "Boundary Spectral Clustering" is not explained clearly.

2. It is crucial to follow up on all datasets of the major baselines (such as Transolver) for proper comparison. Datasets like Elasticity and Plasticity are not included.

3. The proposed technique is not evaluated for zero-shot generalization on discretization (of the domain), meaning that the evaluation samples contain more or fewer points than the training samples.

4. Table 9 indicates that the proposed model requires almost twice the time compared to Transolver.

5. The results in Table 6 are not interpretable. It would be beneficial to add a single performance metric to clearly demonstrate its dependence on $m_I$ and $m_B$. Additionally, it is generally expected that as $m_I$ and $m_B$ increase, the model's performance will improve. However, in many cases, this is not true, and this crucial issue remains unaddressed.

---

> ### Author Rebuttal · Authors · 2025-07-31
>
> ### **Q1: Further clarification on "Boundary Spectral Clustering"  paragraph.**
>
> We apologize for that, and we provide additional implementation details here, which will be included in the appendix of the revised manuscript.
> The underlying process of spectral clustering [1] can be summarized in three main stages (clusters = k):
>
> **Step 1: Constructing the affinity matrix.** Given the input data points $(x_1, x_2, x_3, \dots, x_n)$, the algorithm begins by treating all points as nodes in a graph. The similarity between these nodes is then quantified and stored in an affinity matrix $W$ by k-nearest neighbor (KNN) method. The process is as follows:
>
> $w_{ij} = w_{ji} = \exp \left( - \|| x_i - x_j \||^2 / (2 \sigma^2) \right) \text{ if } x_i \in \text{KNN}(x_j) \text{ or } x_j \in \text{KNN}(x_i), \text{ otherwise } 0.$
>
>
>
> **Step 2: Constructing the Laplacian matrix.** We use the normalized Laplacian matrix, defined as follows:
>
> $
> L=I-D^{-1/2}WD^{-1/2},
> $
>
> where $I$ is the identity matrix, $W$ is the affinity matrix, and $D=\text{diag}(d_1,...,d_n)$ , $d_i=\sum_{j=1}^nw_{i,j}.$
>
> **Step 3: Computation of eigenvectors and clustering completion.** First, compute the first $k$ eigenvectors $(\mathbf{u}_1, \cdots, \mathbf{u}_k)$ of the normalized Laplacian matrix $L$, forming the eigenvector matrix $U \in \mathbb{R}^{n \times k}$. Second, denote $U = ( \mathbf{y}_1, \cdots, \mathbf{y}_n )^{\top}$, where each $\mathbf{y}_i \in \mathbb{R}^k$ corresponds to the $i$ th row of $U$. Third, cluster the points $\mathbf{y}_i (i=1, \cdots, n)$  in $\mathbb{R}^k$ with the k-means algorithm into clusters $C_1, \cdots, C_k$.
>
> We adopt the standard SpectralClustering function from the scikit-learn library:
> SpectralClustering(n\_clusters=clusters,affinity=’nearest\_neighbors’,random\_state=42).
>
>
> ### **Q2: Comparison on Elasticity and Plasticity datasets.**
>
> We extend our evaluation to the Elasticity and Plasticity datasets, as shown in Table R2-1.
> For the Elasticity, the void regions of the shape are treated as complex boundaries for spectral clustering, and $m_I = m_B = 8$.
> For the Plasticity, the four edges of the rectangular region are treated as boundaries for spectral clustering, and $m_I = 4, m_B = 16$.
> We set the number of patches to 64, consistent with the configuration used in Transolver.
> The results are presented in Table R2-1, and our method outperforms all other models.
>
> Table R2-1. Performance comparison (Relative L2 error) on Elasticity and Plasticity datasets.
> | Model| Elasticity | Plasticity |
> | :---: | :---: | :---: |
> | LSM | 0.0218 | 0.0025 |
> | GALERKIN | 0.0240 | 0.0120 |
> | OFORMER | 0.0183 | 0.0017 |
> |  GNOT| 0.0086  | 0.0336 |
> | ONO |0.0118  | 0.0048 |
> |  Transolver| 0.0064 | 0.0012|
>  |  SpiderSolver (Ours)| **0.0061** |  **0.0011** |
>
> Our method has now been evaluated on a total of seven benchmarks.
> Across all of these, SpiderSolver achieves state-of-the-art performance.
> This comprehensive evaluation demonstrates the generality and broad applicability of our approach across diverse PDE systems and geometric settings.
>
> ### **Q3: Lacks evaluation on generalization to domains with different discretization resolutions.**
>
> We have conducted new experiments to evaluate performance under different discretization resolutions.
> Lower-resolution inputs are obtained by randomly downsampling the original point cloud, while higher-resolution inputs are generated via bilinear interpolation. The results are summarized in Table R2-2.
>
> Table R2-2. Model evaluation (relative $L_2$ error) with different discretization resolutions.
>
> | Sampling Rate | Resolutions |Transolver |  SpiderSolver |
> | :---: |  :---: | :---: | :---: |
> |  1/4|   414 |  0.0562  |  0.0446   |
> |  1/2 |  828  |   0.0489|   0.0390  |
> |  1 |   1656  (Original)  |   0.0438   | 0.0322  |
> |  2 |   3312  |  0.0892  |  0.0790  |
> |  4  |   6624  |  0.1390  |  0.1023  |
>
> SpiderSolver outperforms Transolver under both settings, demonstrating stronger generalization to resolution changes. However, both models exhibit performance limitations when applied to the upsampled high-resolution, indicating that generalization to fine-grained discretization remains a challenging direction for future work.
>
> ### **Q4: Proposed model requires almost twice the time compared to Transolver.**
>
> This limitation will be explicitly noted in the revised manuscript.
> To demonstrate the broader value of our method, we provide the following two perspectives.
>
> ***Inference efficiency and scientific utility.***
> It is important to contextualize this overhead relative to traditional numerical solvers. For example, a conventional $k$–$\varepsilon$ turbulence simulation typically takes around 50 minutes for a single prediction of the Shape-Net Car. In contrast, both Transolver and SpiderSolver produce results within tens of milliseconds, achieving a speedup of over three orders of magnitude. From the perspective of accelerating scientific simulations, millisecond-level inference remains highly practical for deployment.
>
> ***State-of-the-Art performance on seven PDE datasets.***
> SpiderSolver has been validated across seven diverse benchmarks. Across all cases, it achieves state-of-the-art performance. This broad evaluation underscores the generality and practical value of our method, even when accounting for its modest increase in inference time.
>
> ### **Q5: No clear metric shows how  $m_I, m_B$ affect performance; results are inconsistent and hard to interpret.**
>
> We observe no clear trend in model performance across different values of $m_I$ and $m_B$ in Table 6.
> We attribute this to the large number of patches used, which results in too few discrete points within each patch, which is insufficient to capture the underlying physical field characteristics.
> To validate this hypothesis, we conducted additional experiments using fewer patches.
> As shown in Tables R2-3 and R2-4, our model's performance improves as the number of patches increases.
>
> Table R2-3. Ablation study of SpiderSolver with varying $m_I$ and $m_B$ on AirfRANS.
>
> | $(m_I, m_B)$| Patch Number ($m_Bm_I+ m_B$)  | Vol ↓ |Surf ↓|$C_L$ ↓ |  $\rho_L$ ↑ |
> |---|---|----|---|---|---|
> |(2,2)|6| 0.0026 | 0.0130 |  0.0885 | 0.9987 |
> |(4,2) |10|   0.0021 | 0.0101 | 0.0818|0.9988 |
> |(2,4) |12 | 0.0019 |0.0064 | 0.0751|0.9989 |
> |(6,2)| 14 | 0.0020  |0.0079 |0.0624 | 0.9990  |
> |(2,6) |18|  0.0019 |0.0075 | 0.0583 | 0.9988 |
>
>
> Table R2-4. Ablation study of SpiderSolver with varying $m_I$ and $m_B$ on Blood Flow.
>
> |$(m_I, m_B)$|Patch Number ($m_Bm_I+ m_B$) | Vol ↓|
> |-----|---|----|
> |(2,2) | 6 | 0.0372 |
> |(4,2) | 10 |0.0366 |
> | (2,4) | 12  | 0.0359  |
> | (6,2) | 14 | 0.0352 |
> |(2,6) | 18 | 0.0350 |
>
> ### **Q6: Clarification on the discrepancy between Table 5 and Table 3**
>
> We summarize the distinct purposes and experimental setups of Table 5 and Table 3.
>
> ***Clarification on Table 3 and Table 5 in our manuscript.***
> We believe the reviewer may be referring to Table 3 and Table 5 in our manuscript, as we did not find any direct connection between Tables 3 and 5 in the original Transolver paper.
>
> ***Different experimental settings.***
> Table 3 employs distinct training and test sets tailored for out-of-distribution (OOD) experiments, which differ from the standard in-distribution setup used in Table 5, including extrapolation with respect to Reynolds number and angle of attack.
> The corresponding experimental setup is detailed in Lines 321–328 on page 8 of our manuscript.
> Table 5 reports the results of our ablation studies (Lines 341-348 on page 9).
>
> ### **Q7: Instance-wise boundary clustering may help, consider ablation.**
>
> We provide a detailed explanation of the limitations of instance-specific boundary clustering, followed by an ablation study evaluating its effect.
>
> ***Limitations of instance-specific boundary clustering.***
> As shown in Figure 4(b) and Figure 10 in Appendix A, performing boundary clustering independently for each instance results in unaligned clustering. That is, corresponding regions across different instances (e.g., the rooftops of different cars) are not guaranteed to be assigned to the same cluster. This lack of alignment leads to inconsistencies in how the interior domain is partitioned into patches, making it difficult for the attention mechanism to capture spatial correspondences across samples. This observation motivates our use of a template shape (Lines 153–165 on page 5).
>
> ***Ablation with instance-specific clustering.***
> To further assess this alternative, we conducted additional experiments where boundary clustering was performed independently for each instance, and all other components of our model remained unchanged, as shown in Tables R2-5 and R2-6.
>
> Table R2-5. Effect of instance-specific clustering on model performance on Shape-Net Car.
>
> | Model| Vol ↓| Surf ↓ | $ C_D $ ↓ | $ \rho_D $ ↑ |
> |---|--|---|---|--|
> |SpiderSolver (Ours)| 0.0210 | 0.0738 | 0.0100| 0.9928 |
> |Instance-specific clustering | 0.0222 | 0.0751|0.0124 | 0.9911|
>
> Table R2-6. Effect of instance-specific clustering on model performance on AirfRANS.
>
> | Model | Vol ↓ | Surf ↓ | $C_L$  ↓ | $\rho_L$ ↑ |
> |---|---|---|---|----|
> | SpiderSolver (Ours)|0.0017| 0.0043 | 0.0741| 0.9988|
> | Instance-specific clustering|0.0022|0.0099 |0.1013| 0.9986|
>
> As shown in Tables R2-5 and R2-6, instance-wise boundary clustering leads to a performance drop. This degradation arises from misaligned clustering results across different instances, which in turn causes geometric inconsistency in the partitioning of interior points within the PDE domain.
>
> &nbsp;
>
> [1] Von Luxburg U. A tutorial on spectral clustering [J]. Statistics and computing, 2007, 17(4): 395-416.

---

### Official Review · Reviewer_Rhww · 2025-07-03

**Clarity:** 3
**Significance:** 3
**Originality:** 3
**Rating:** 4
**Confidence:** 3

**Summary:**

This paper introduces SpiderSolver, a novel deep learning model designed to solve PDE on domains with complex geometries and irregular discretizations. The key innovation is a geometry-aware transformer architecture that addresses the challenges of applying standard transformers to such problems.

The technical contributions include:
1. The proposed method first uses spectral clustering on the boundary points of the domain to create shape-aware partitions. Then, it divides the interior domain into "spiderweb-like" patches based on these boundary clusters and the distance of interior points from the boundary. This tokenization is designed to be more physically intuitive and computationally efficient than methods that treat all mesh points individually.

2. To ensure consistent tokenization across different instances of a similar geometry (e.g., various car shapes), the authors propose an alignment strategy. They use optimal transport to match the boundary points of any given shape to a pre-defined template shape, transferring the clustering from the template to the new instance.

3. SpiderSolver transformer employs a bi-level attention system to understand the coarse-grained and fine-grained features.

The authors evaluate SpiderSolver on five diverse datasets, including simulations based on Shape-Net Car/AirfRANS, blood flow in the human aorta, and canonical Navier-Stokes and Darcy flow problems. The results demonstrate that SpiderSolver consistently achieves SOTA performance, outperforming previous neural operators and transformer-based PDE solvers, particularly in its generalization to OOD cases.

**Questions:**

Please address the weakness points I mentioned.

**Ethical Concerns:**

["NO or VERY MINOR ethics concerns only"]

**Final Justification:**

Many thanks for providing further dicussions and results. They fully addressed my concerns. I will keep my score.

**Limitations:**

Please address the weakness points I mentioned.

**Quality:**

3

**Strengths And Weaknesses:**

S1. The core idea of "spiderweb tokenization" is a significant and intuitive contribution.
S2. The dual-level attention mechanism is an intuitive yet effective way to mitigate the quadratic complexity of standard transformers.
S3. The use of optimal transport for aligning boundary clustering is a sophisticated solution to handle variations in shape within a class of objects.

W1. The authors acknowledge that the current formulation of SpiderSolver relies on a well-defined, closed boundary. This limits its applicability to problems with open-boundary domains or domains with multiple, disconnected internal boundaries that are not part of a single, continuous surface. While the Bounded Navier-Stokes dataset has multiple obstacles, the tokenization still relies on the minimum distance to any boundary component, which may not be optimal for capturing interactions between the obstacles themselves.

W2. The aligned clustering method requires a "template shape" for each class of geometry. The paper does not fully detail how this template is chosen or how sensitive the model's performance is to this choice. An arbitrary or poorly chosen template could potentially lead to suboptimal tokenization for the entire dataset.

W3. While the paper notes that spectral clustering is computationally manageable for the datasets used (e.g., around 3,600 surface points for the car), this treatment could become a bottleneck for problems with extremely high-resolution boundary meshes containing millions of points.

---

> ### Author Rebuttal · Authors · 2025-07-31
>
> ### **Q1: Reduced suitability for open or multi-boundary domains, not optimal for obstacle interaction modeling.**
>
> We first clarify the applicability of SpiderSolver to open and multi-boundary domains, and then explain how obstacle interactions are already modeled within our framework. Finally, we conducted ablation studies to support our claims.
>
> ***(1) Clarification on the applicability of SpiderSolver to boundary conditions.***
> We acknowledge that SpiderSolver is currently not applicable to PDEs with open-boundary domains, as stated in 'Limitations and future work'  in manuscript.
> However, it does support domains with multiple disconnected internal boundaries, such as those present in the Bounded Navier-Stokes dataset.
>
> ***(2) Obstacle Interaction Analysis.***
> SpiderSolver performs spectral clustering on the surfaces of all obstacles, producing $m_B$ boundary clusters. The interior domain is then partitioned into $m_B m_I$ subregions, resulting in a total of $m_B m_I  + m_B$ tokens.
> **Token-level attention captures obstacle interactions:** Coarse-grained Attention in SpiderSolver naturally captures interactions between the obstacles, as the $m_B m_I + m_B$ tokens attend to each other directly.
> **Shape-aware partitions reflect obstacle geometry:**  As shown in Figure 14 (page 15), the interior partitions closely align with the geometry of the obstacles. These shape-aware partitions ensure that the learned attention maps inherently encode the mutual influence among different obstacles.
>
> ***(3) Ablation of merging obstacle surface tokens.***
> We conduct an ablation study in which the $m_B$ tokens corresponding to obstacle surfaces are merged into a single token, resulting in a total of $m_B m_I  + 1$ tokens. The results are reported in Table R1-1. The experimental results indicate that explicitly modeling interactions between obstacles leads to improved model performance.
>
> Table R1-1. Effect of obstacle surface token merging on prediction accuracy.
>
> | Methods  | Relative $L_2$ error ↓ |
> |----|---|
> | SpiderSolver with merged obstacle surface tokens  | 0.0432 |
> | SpiderSolver (Ours)  | 0.0376 |
>
> ###  **Q2: Template selection is unclear, with potential sensitivity and suboptimal tokenization arising from poor template choices.**
>
> We first clarify the selection of the reference shape and the construction of the template shape, highlighting the distinction between the two.
> We then performed an ablation study with different reference shapes.
>
> ***Reference shape and template shape.***
> The reference shape is randomly selected from the training data, and the template shape is then obtained by averaging the points of all training shapes.
> To enable meaningful pointwise averaging, all shapes are first aligned to the reference shape using optimal transport.  The reference shape serves only to provide a consistent indexing, and it does not affect the geometry of the resulting template shape.
>
> ***Ablation on reference shape selection.***
> As shown in  Table R1-2, our model exhibits strong robustness to the choice of reference shape on the Shape-Net Car, consistently outperforming Transolver across different reference shapes in Table R1-2.
>
> Table R1-2.  Ablation study on reference shape selection for Shape-Net Car.
>
> | Methods | Vol ↓ | Surf ↓ | $ C_D $ ↓ | $ \rho_D $ ↑ |
> |----|---|--|----|---|
> | Reference shape 1  | 0.0215  | 0.0730  |  0.0095  | 0.9934   |
> | Reference shape 2  |  0.0212 | 0.0721 | 0.0103 |  0.9935 |
> | Reference shape 3  |0.0215   | 0.0725| 0.0105 | 0.9921 |
> | Reference shape 4  |  0.0210 | 0.0727 | 0.0103 | 0.9934 |
> | Results in manuscript  |   0.0210  | 0.0738 | 0.0100  | 0.9928   |
> | **Mean** |  0.0212  |  0.0728  |  0.0101  | 0.9930  |
> | **Standard Deviation** |  0.0002  |  0.0006  |  0.0003 | 0.0005  |
> | Transolver |  0.0228  |  0.0793  | 0.0129 | 0.9916 |
>
> ***Measuring clustering consistency under different reference shapes.***
> As shown in Table R1-3,
> Dice and IoU are used to evaluate the consistency of spectral clustering results of five template shapes based on five different reference shapes in Table R1-2. Dice and IoU are computed based on the intersection and union of the five clusterings.
>
> Table R1-3. Measuring clustering consistency under five reference shapes.
> | Case|  Dice ↑ | IoU ↑ |
> |:---:|:---:|:---:|
> | Five clusterings | 0.9975  |0.9951 |
>
> ###  **Q3: Computational cost of spectral clustering on high-resolution meshes.**
>
> We first describe the current strategies employed in our model to reduce the computational cost of spectral clustering. Secondly, we add the model training with subsampling-guided clustering.
> Finally, we outline our planned solution for efficiently handling boundary with millions points in future extensions.
>
> ***(1) Reducing spectral clustering cost in our current model.***
> Our model incorporates two strategies to reduce spectral clustering cost: template-based transfer to avoid per-instance clustering, and a sparse Laplacian decomposition using k-nearest neighbor affinity.
>
> **Single clustering via template transfer:** During training, spectral clustering is performed only once on the template shape (i.e., average shape of aligned geometric category), rather than on every individual instance. The clustering results are then transferred to other individual shapes using optimal transport. No spectral clustering is needed during inference.
>
> **Efficient spectral decomposition with neighbor-based affinity:**
> We adopt an efficient spectral clustering method based on a k-nearest neighbor graph, and only the top $m_B$ eigenvectors (typically $m_B=2 \sim 8$) of the Laplacian matrix are computed.
> Table R1-4 compares our clustering approach with the classical radial basis function kernel
> (RBF-based) method [1], which constructs a fully connected affinity matrix based on pairwise distances.
>
> Table R1-4. Comparison for spectral clustering on a car surface with 6 clusters.
> | Methods | Number of Points | Memory Usage (MB)  ↓| Time (s)  ↓|
> | :---: | :---: | :---: | :---: |
> |Classical RBF| 3586 |551.90 | 3.06 |
> |Our approach| 3586 |138.10 | 0.26 |
>
> **Clustering overhead analysis:** Table R1-5 provides a clearer assessment of the computational overhead.
> The original car surface contains 3,586 points. Higher-resolution surfaces were generated using the feature-preserving edge-aware resampling (EAR) algorithm, while lower-resolution surfaces were obtained via standard uniform random sampling.
>
> Table R1-5. Computational cost of spectral clustering under different surface resolutions.
>
> | Sampling Rate | Number of Points | Memory Usage (MB)  ↓| Time (s)  ↓|
> | :---: | :---: | :---: | :---: |
> |1/8 | 448 | 132.90 | 0.04 |
> | 1/6 | 597 | 133.80 | 0.05 |
> | 1/4 | 896 | 133.91 | 0.07 |
> | 1/2 | 1793 | 134.57 | 0.15 |
> | 1 | 3586 (Original) | 138.10 | 0.26 |
> | 2 | 7172 | 146.57 | 1.04 |
> | 4 | 14344 | 168.00 | 4.54 |
> | 6 | 21516 | 188.73 | 9.50 |
> | 8 | 28688 | 212.23 | 20.74 |
> | 16 | 57376 | 297.82 | 90.39 |
> | 32 | 114752 | 468.07 | 373.37 |
>
> As shown in Table R1-5, clustering becomes time-consuming at high resolutions.
> To address this, we propose a subsampling-based clustering strategy and conduct experiments to verify its effectiveness.
>
> ***(2) Speeding up the spectral clustering on larger scale boundary point clouds by  subsampling-based spectral clustering.***
>
> To efficiently speed up the spectral clustering on large scale boundary point cloud, we can first subsample the original point cloud and compute the spectral clustering on the subsampled boundary point cloud, then upsample the clustering by labeling the remaining points in the original point set  using k-nearest neighbor (KNN) classifier.  To evaluate this idea, we use the edge-aware resampling (EAR) algorithm to generate high-resolution data on Shape-Net Car. Then the
> spectral clustering is performed on the original 3586 points, and the resulting cluster labels are propagated to the high-resolution points via the KNN classifier.
> We compare the propagated clustering with the clustering of directly performing spectral clustering on the high-resolution point cloud, as shown in Table R1-6.
> The subsampling-based clustering yields results that are highly consistent with those obtained from direct clustering at high resolution, while substantially reducing the computational time.
> The total clustering time under the subsampling-based strategy consists of the spectral clustering time on the original 3586 points (0.26s), plus the time required for label propagation via the KNN classifier (Table R1-6).
>
> Table R1-6. Clustering quality under point cloud subsampling at different resolutions.
>
> | Resolutions | Dice ↑ | IoU  ↑ |Time of KNN Classifier (s) ↓|
> | :---: | :---: | :---: | :---: |
> | 7172  |0.9762|0.9521|   0.0054 |
> | 14344  |0.9678|0.9393| 0.0125 |
> | 21516  | 0.9621|0.9278| 0.0207 |
> | 28688  | 0.9613|0.9257| 0.0277|
> | 57376  | 0.9478|0.9013|  0.0575|
> | 114752  | 0.9324|0.8748| 0.1226 |
>
> **Effect on the performance under subsampling-based spectral clustering:**
> We futher evaluate the effect of subsampling-based spectral clustering on the final performance. On the Shape-Net Car, we perform the subsampling-based spectral clustering on the subsampled car surface of 896 points (i.e., one-fourth of the original 3586 resolution).  We then train the model using this approximate clustering, as shown in Table R1-7. The model exhibits only a slight performance degradation under this setting, while still outperforming Transolver, demonstrating the effectiveness of our subsampling-based clustering strategy.
>
> Table R1-7. Effect of subsampling-based clustering on model performance.
>
> | Methods    | Vol ↓ | Surf ↓ | $ C_D $ ↓ | $ \rho_D $ ↑ |
> |-----|---|---|--|--|
> | Transolver | 0.0228 | 0.0793  | 0.0129 | 0.9916 |
> | SpiderSolver (original) | 0.0210  | 0.0738 | 0.0100 | 0.9928 |
> | SpiderSolver with subsampling-based clustering | 0.0213  |0.0742 | 0.0107 | 0.9924|
>
> [1] Ng A, Jordan M, Weiss Y. On spectral clustering: Analysis and an algorithm. NeurIPS 2001.

---

> > ### Author Response · Authors · 2025-08-08
> >
> > We sincerely thank the reviewer for  insightful comments. In response, we have provided detailed ablation studies, clarified the applicability of our method to multi-boundary domains, demonstrated its robustness to template selection, and studied a subsampling-based strategy to address spectral clustering overhead on high-resolution meshes.  We will incorporate all these clarifications and experimental results into our paper.
> >
> > We would appreciate hearing from you to confirm whether these responses have addressed your concerns. We are also fully open to addressing any remaining questions you may have.

---

### Decision · Program_Chairs · 2025-09-17

**Decision:**

Accept (poster)

**Comment:**

The paper addresses the challenging problem of solving PDEs on complex geometries with a geometry-aware transformer architecture. Reviewers uniformly found the work to be novel and timely, praising its strong motivation, solid technical design, and clear connections between geometric priors and the model structure. The empirical results, though limited in scale, demonstrate consistent improvements over competitive baselines and support the claimed advantages. The rebuttal further clarified technical points and addressed questions on scalability and robustness to the reviewers’ satisfaction. Overall, given the positive feedback across the board and the clear contributions in both methodology and application, I recommend acceptance.